# Zero-shot Visual Relation Detection via Composite Visual Cues from Large Language Models

**Lin Li**[1,2], **Jun Xiao**[1], **Guikun Chen**[1], **Jian Shao**[1], **Yueting Zhuang**[1], **Long Chen**[2]*

[1]Zhejiang University    [2]The Hong Kong University of Science and Technology

{mukti, junx, guikun.chen, jshao, yzhuang}@zju.edu.cn, longchen@ust.hk

https://github.com/HKUST-LongGroup/RECODE

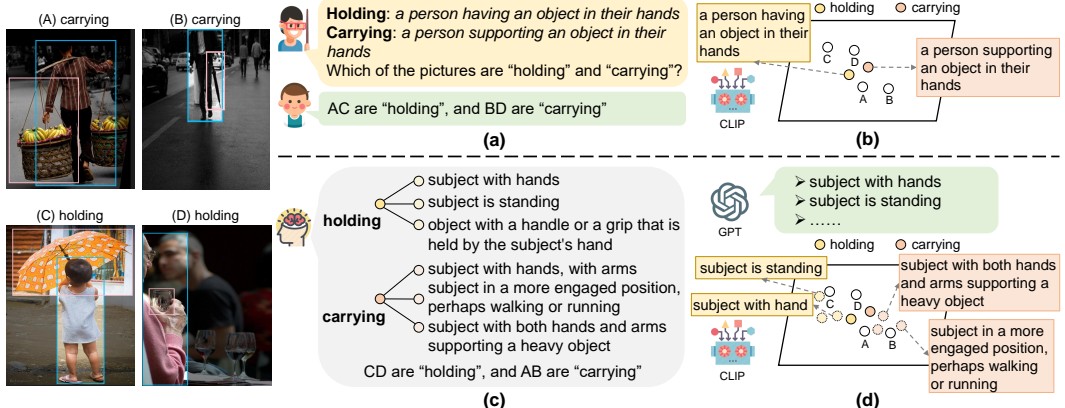

Figure 1: Illustration of the challenges of VRD with similar relation categories **holding** and **carrying**. Four images and their ground-truths are on the left. The subject and object for each triplet are denoted by blue and pink boxes, respectively. **(a)** A child may incorrectly identify these two relations only based on similar concepts alone. **(b)** Using class-based prompts, CLIP always maps these two relations to adjacent locations in the semantic space. **(c)** We humans always utilize composite visual cues to correctly distinguish between different relations. **(d)** Our proposed RECODE uses LLM (*e.g.*, GPT) to generate composite descriptions that aid the CLIP model in distinguishing between them.

## Abstract

Pretrained vision-language models, such as CLIP, have demonstrated strong generalization capabilities, making them promising tools in the realm of zero-shot visual recognition. Visual relation detection (VRD) is a typical task that identifies relationship (or interaction) types between object pairs within an image. However, naively utilizing CLIP with prevalent *class-based* prompts for zero-shot VRD has several weaknesses, *e.g.*, it struggles to distinguish between different fine-grained relation types and it neglects essential spatial information of two objects. To this end, we propose a novel method for zero-shot VRD: **RECODE**, which solves RElation detection via COmposite DEscription prompts. Specifically, RECODE first decomposes each predicate category into subject, object, and spatial components. Then, it leverages large language models (LLMs) to generate description-based prompts (or visual cues) for each component. Different visual cues enhance the discriminability of similar relation categories from different perspectives, which significantly boosts performance in VRD. To dynamically fuse different cues, we further introduce a chain-of-thought method that prompts LLMs to generate rea-

---

*Long Chen is the corresponding author. Work was done when Lin Li visited HKUST.

37th Conference on Neural Information Processing Systems (NeurIPS 2023).

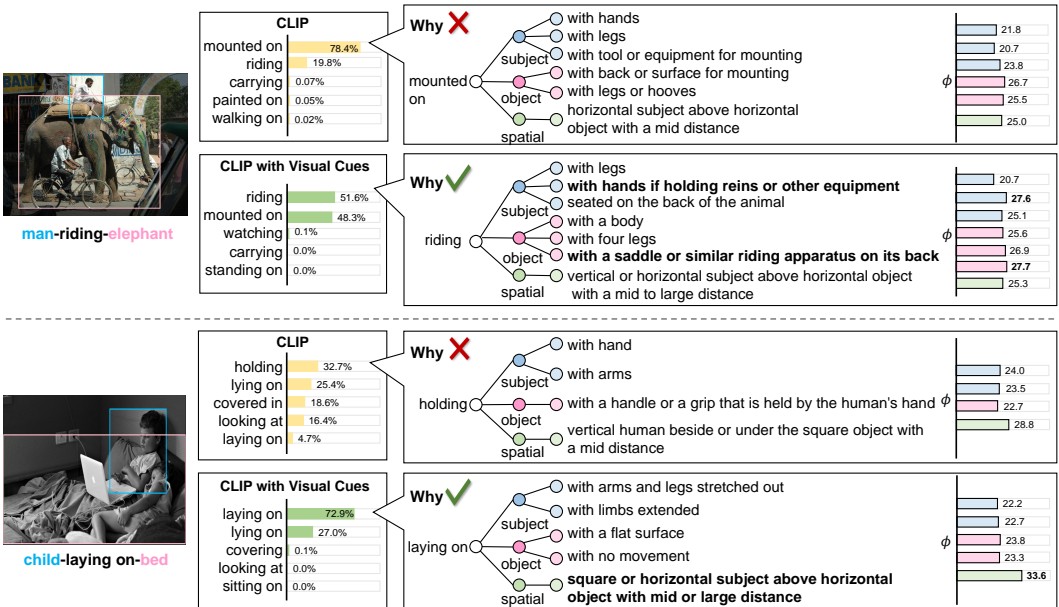

Figure 2: A comparative analysis of predictions made by RECODE and baseline CLIP using class-based prompts. It illustrates how our method offers interpretability to the relation classification results through the similarity $\phi$ between the image and the description-based prompts.

sonable weights for different visual cues. Extensive experiments on four VRD benchmarks have demonstrated the effectiveness and interpretability of RECODE.

# 1 Introduction

Recent advances in pretrained vision-language models (VLMs) [1, 2, 3, 4] (*e.g.*, CLIP [1]), have shown remarkable generalization ability and achieved impressive performance on zero-shot recognition tasks. Specifically, CLIP employs two encoders: an image encoder that converts images into visual features, and a text encoder that transforms sentences into semantic features. This design allows the encoders to map different modalities into a common semantic space. When the inputs to the text encoder are *class-based prompts*, such as "A [CLASS]", "A photo of [CLASS]", CLIP can compare the image and prompts in the shared semantic space, thereby enabling zero-shot recognition of novel categories [1]. Compared to object recognition, visual relation detection (VRD) is much more challenging, which needs to identify the relation types between object pairs within an image in the form of ⟨subject, relation, object⟩ [5, 6, 7, 8, 9]. It differs from object recognition in that it requires an understanding of how objects are related to each other. By crafting class-based prompts to describe these relation types, CLIP could potentially be extended to perform zero-shot VRD.

However, this straightforward baseline presents notable challenges. Imagine you are a child asked to distinguish relation categories "holding" and "carrying", both involving a person and an object. Based on the similar concepts of "holding" (*i.e.*, a person having an object in their hands) and "carrying" (*i.e.*, a person supporting an object in their hands), it would be difficult to determine the correct prediction (*cf.*, Figure 1(a)). In other words, class-based prompts of "holding" and "carrying" might be projected to adjacent locations in semantic space by CLIP, leading to a **relation sensitivity** issue: CLIP struggles to differentiate between the subtle nuances of similar relations. Secondly, class-based prompts overlook the unique spatial cues inherent to each relation category, leading to a **spatial discriminability** issue. The "holding" category generally suggests the object being at a certain height and orientation relative to the person, while "carrying" implies a different spatial position, typically with the object located lower and possibly supported by the person's entire body. The neglect of spatial cues leads to inaccuracies in distinguishing between such spatial-aware relation categories. Moreover, applying CLIP in this manner brings about a **computational efficiency** issue. Using CLIP requires cropping each union region of a subject-object pair separately from the original image (*i.e.*, $N^2$ crops for $N$ proposals), leading to computational inefficiencies.

Nonetheless, we humans can distinguish relation categories from different visual cues. For example, from the subject's perspective, we could think that in the case of "holding", a person might be standing while having an object, such as an umbrella, in their hand. Meanwhile, in the case of "carrying", a person should be in a more engaged position, perhaps walking or running with both hands and arms supporting a heavy object, like a suitcase. In addition, spatial cues also play an important role in identifying these relation categories. For example, when a person is carrying an umbrella, the umbrella is usually positioned lower and closer to the person's body compared to when the person is holding an umbrella. Based on these visual cues, we can easily identify scenarios such as "`person-holding-umbrella`" and "`person-carrying-umbrella`" as in Figure 1(c).

Inspired by our humans' ability to extract and utilize different visual cues, we present a novel method for zero-shot VRD: **RECODE**, which classifies RElation via COmposite DEscriptions. It first uses large language models (LLMs) [10], to generate detailed and informative descriptions[2] for different components of relation categories, such as subject, object, and spatial. These descriptions are then used as description-based prompts for the CLIP model, enabling it to focus on specific visual features that help distinguish between similar relation categories and improve VRD performance. Specifically, for the subject and object components, these prompts include visual cues such as appearance (*e.g.*, with leg), size (*e.g.*, small), and posture (*e.g.*, in a sitting posture). For the spatial component, these prompts include cues related to the spatial relationships between objects, such as relative position and distance. By incorporating different visual cues, RECODE enhances the discriminability of similar relation categories, such as "riding" and "mounted" based on the different postures of the subject, *e.g.*, "seated on the back of animal" for the subject of "riding". Similarly, spatial visual cues can be used to differentiate between "laying on" and "holding" based on the relative position between the subject and object, such as "subject above object" and "subject under object" (*cf.*, Figure 2).

In addition, we explore the limitations of several description generation prompts for visual cue, *e.g.*, relation class description prompt [11], and then design a guided relation component description prompt that utilizes the high-level object categories to generate more accurate visual cues for each relation category. For instance, if the high-level category of object is "animal", the generated object descriptions for relation "riding" are tailored to the "animal" category, *e.g.*, "with four legs", instead of the "product", *e.g.*, "with wheels". Meanwhile, to better fuse the evidence from different visual cues, we further leverage LLMs to predict reasonable weights for different components. Particularly, we design a chain-of-thought (CoT) method [12] to break down this weight assignment problem into smaller, more manageable pieces, and prompt LLM to generate a series of rationales and weights.

To evaluate our RECODE, we conducted experiments on four benchmark datasets: Visual Genome (VG) [13] and GQA [14] datasets for scene graph generation (SGG), and HICO-DET [15] and V-COCO [16] datasets for human-object interaction (HOI) detection. Experimental results prove the generalization and interpretability of our method. In summary, we made three main **contributions** in this paper: 1) We analyze the weaknesses of the prevalent class-based prompt for zero-shot VRD in detail and propose a novel solution RECODE. RECODE leverages the power of LLMs to generate description-based prompts (visual cues) for each component of the relation class, enhancing the CLIP model's ability to distinguish between various relation categories. 2) We introduce a chain-of-thought method that breaks down the problem into smaller, more manageable pieces, allowing the LLM to generate a series of rationales for each cue, ultimately leading to reasonable weights for each component. 3) We conduct experiments on four benchmark datasets and demonstrate the effectiveness and interpretability of our method.

## 2 Approach

Typically, VRD is comprised of two sub-tasks: object detection and relation classification [5]. Since zero-shot object detection has been extensively studied [17, 1, 11], in this paper, we primarily focus on zero-shot relation classification. Specifically, given the bounding boxes (bboxes) $\{b_i\}$ and object categories $\{o_i\}$ of all objects, our target is to predict the visual relation (or predicate/interaction) categories $\{r_{ij}\}$ between pairwise objects. To facilitate presentation, we use $s$, $o$, and $p$ to denote the subject, object, and their spatial position in a triplet respectively, and $r$ to denote the relation category.

**Class-based Prompt Baseline for Zero-Shot VRD.** Following recent zero-shot object recognition methods, a straightforward solution for zero-shot VRD is the CLIP with class-based prompt. Specif-

---

[2]We use a description to represent a visual cue.

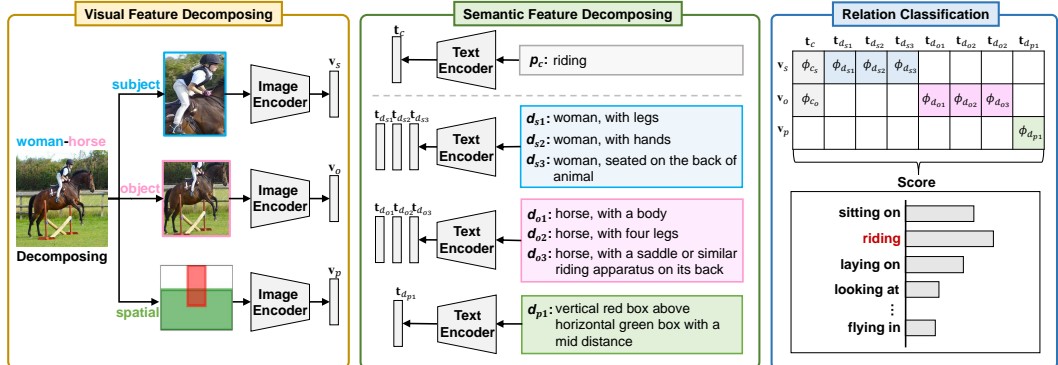

Figure 3: The framework of RECODE. 1) *Visual feature decomposing* decomposes the triplet into subject, object, and spatial features. 2) *Semantic feature decomposing* decomposes relation categories into subject, object, and spatial descriptions. 3) *Relation classification* calculates similarities between decomposed visual and semantic features and applies softmax to obtain the probability distribution.

ically, a pretrained CLIP image encoder $V(\cdot)$ and a pretrained CLIP text encoder $T(\cdot)$ are used to classify pairwise objects with a set of relation classes. For each relation class, a natural language **class-based prompt** $p_c$ is generated, incorporating the relation information, *e.g.*, "[REL-CLS]-ing/ed" or "a photo of [REL-CLS]". Each prompt is then passed through $T(\cdot)$ to get semantic embedding $t$, while the union region of a subject-object pair is passed through $V(\cdot)$ to get visual embedding $v$. The cosine similarity between $v$ and $t$ of different relation categories is calculated and processed by a softmax function to obtain the probability distribution over all relation categories.

## 2.1 Zero-shot VRD with Composed Visual Cues

To overcome the limitations of class-based prompts, we propose a novel approach RECODE for zero-shot VRD. It consists of three parts: visual feature decomposing, semantic feature decomposing, and relation classification (*cf.*, Figure 3). In the first two parts, we decompose the visual features of the triplet into subject, object, and spatial features, and then generate semantic features for each component. In the last part, we calculate the similarities between the decomposed visual features and a set of semantic features, and aggregate them to get the final predictions over all relations.

**Visual Feature Decomposing.** To enhance spatial discriminability and computational efficiency, we decompose the visual features of a triplet into subject, object, and spatial features. *For subject and object features*, we crop the regions of the subject and object from the original image using the given bboxes $b_s$ and $b_o$, and encode them into visual embeddings $v_s$ and $v_o$ using the image encoder $V(\cdot)$ of CLIP. *For spatial features*, we aim to obtain the spatial relationship between the subject and object based on their bounding boxes. However, directly obtaining all spatial images based on the given bounding boxes is computationally expensive due to the diversity of spatial positions ($N^2$ each image). To address this, we simulate the spatial relationship between the subject and object using a finite set of spatial images, represented by red and green bboxes respectively. We define four attributes (shape, size, relative position, and distance) based on bounding box properties. Each attribute is assigned a finite set of values to construct a finite set of simulated spatial images. For a given triplet, we match the calculated attribute values with the most similar simulated image[3]. The matched spatial image is then encoded into a visual embedding $v_p$ using $V(\cdot)$ of CLIP.

**Semantic Feature Decomposing.** To improve the CLIP model's ability to distinguish between different relation classes, we incorporate a set of **description-based prompts** $D$ to augment the original class-based prompt for each relation category. *For the subject and object components*, we generate a set of description-based prompts $D_s$ and $D_o$ to provide additional visual cue information, the generation process is described in Sec. 2.2. These prompts contain object categories with specific visual cues that highlight the unique characteristics of the relation being performed, *e.g.*, "women, with legs", which enhances the discriminability between similar relation categories. *For the spatial component*, it only contains a set of description-based prompts $D_p$ that include information about the relative position and distance between the subject and object in the image. By incorporating

---

[3]Due to the limited space, the details are left in the appendix.

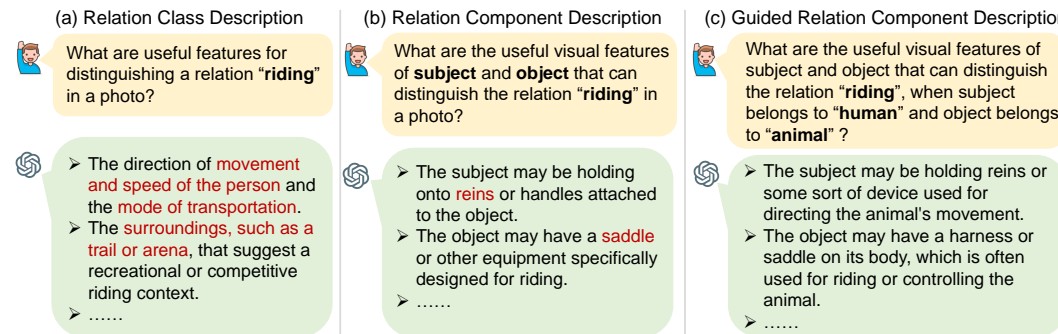

Figure 4: Examples of different prompts used for generating descriptions of visual cues. (a) ***Relation class description*** generates descriptions for each relation class directly. (b) ***Relation component description*** generates descriptions for each component of the relation separately. (c) ***Guided relation component description*** incorporates high-level object category guide generation process.

this additional information, we aim to distinguish between relations based on spatial location. After generating these sets of description-based prompts, we obtain semantic embeddings $\{t_{d_{s_i}}\}$, $\{t_{d_{o_i}}\}$, and $\{t_{d_{p_i}}\}$ using a text encoder $T(\cdot)$, separately. These embeddings, along with the class-based prompt embedding $t_c$, are used for relation classification.

**Relation Classification.** In this step, we compute the similarity score between the visual and semantic features to obtain the relation probability distribution. We first calculate the cosine similarity $\phi(\cdot, \cdot)$ between each visual embedding and semantic embedding for each relation category $r$. The final score incorporates both class-based and description-based prompts, and is calculated as follows:

$$S(r) = \underbrace{\phi(v_s, t_c) + \phi(v_o, t_c)}_{\text{class-based prompts}} + \underbrace{\sum_{k \in \{s,o,p\}} \frac{w_k}{|D_k(r)|} \left[ \sum_{d_{k_i} \in D_k(r)} \phi(v_k, t_{d_{k_i}}) \right]}_{\text{description-based prompts}}, \tag{1}$$

where $w_k$ represents the importance of visual cues for each component $k \in \{s, o, p\}$, and $|D_k(r)|$ denotes the number of visual cues in $D_k(r)$ for relation category $r$. We compute the similarity of individual visual cues for each component and then obtain their average. The weights of different components are determined by a LLM, which will be discussed in Sec. 2.2. Finally, we apply a softmax operation to the scores to obtain the probability distribution over all relation categories.

## 2.2 Visual Cue Descriptions and Weights Generation

LLMs, such as GPT [10], have been shown to contain significant world knowledge. In this section, we present the process of generating descriptions of visual cue $D_s$, $D_o$, and $D_p$, as well as the weights $w_s$, $w_o$, and $w_p$ for each component of each relation category using LLMs.

### 2.2.1 Visual Cue Descriptions

In this section, we explore methods for generating descriptions of visual cues for relation decomposition. Inspired by the work [11] of zero-shot image classification, we first propose **relation class description** prompt, which generates descriptions from the perspective of class-level (*cf.*, Figure 4(a)). It has the advantage of producing descriptions that are easy to interpret and understand. However, it may result in overly diverse and information-rich descriptions that could hinder the extraction of meaningful visual cues, *e.g.*, "speed of the person" in Figure 4(a).

To address this limitation, we then consider another **relation component description** prompt, which involves decomposing the relation into its subject and object components and generating descriptions of their visual features separately (*cf.*, Figure 4(b)). While this type of prompt allows for more focused and specific descriptions of visual cues, it may not be effective in capturing the variations in visual features between different subject-object category pairs. For example, "`man-riding-horse`" and "`person-riding-bike`" typically have totally different visual features for the object. The visual cues "reins" and "saddle" of the object in Figure 4(b) are inappropriate for a "bike".

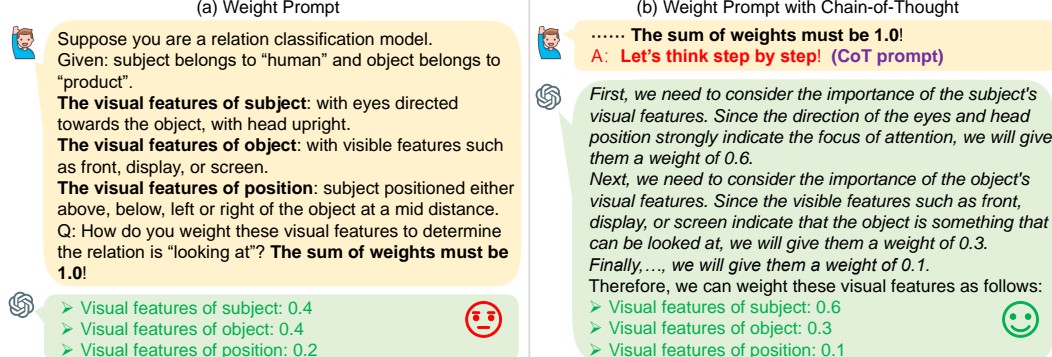

Figure 5: Illustration of the effectiveness of CoT method in generating reasonable visual cue weights. (a) **Prompt without CoT**. LLM assigns same weights for subject and object. (b) **Prompt with CoT**. LLM analyzes the importance of each cue step by step and assigns more reasonable weights.

Therefore, we design **guided relation component description** prompt. It builds upon the second method by incorporating the high-level category information of the object into the generation process, leading to more accurate and informative descriptions of the visual features of both the subject and object components (*cf.*, Figure 4(c)). To achieve this, we classify the object into high-level classes, such as "human", "animal", and "product", to guide the description generation. For example, "bike" is classified as "product", and "horse" is classified as "animal". This allows for the separate generation of visual feature descriptions for each high-level object class, *e.g.*, "a harness or saddle on its body" for "animal", resulting in more precise and relevant visual cues for each relation category[3].

### 2.2.2 Visual Cue Weights

Intuitively, different combinations of visual cues may have varying degrees of importance in relation classification. For example, for relation "looking at", the visual cue "with visible features" of the object may not be as informative as the visual cue "with eye" of the subject. To account for this, we leverage the impressive knowledge and reasoning abilities of LLMs to analyze the discriminative power of different visual cues and dynamically assign weights accordingly. Specifically, we provide each combination of visual cues as input to LLM and prompt it to determine the appropriate weight for each cue for distinguishing the given predicate. The prompts used for this purpose are in Figure 5.

**Chain-of-Thought (CoT) Prompting.** To ensure the generated weights are reasonable, we utilize a CoT method that has demonstrated remarkable reasoning abilities [12, 18]. Specifically, we prompt the LLM to generate rationales by using the stepwise reasoning prompt "**Let's think step by step!**" to break down the problem into smaller, more manageable pieces. Then LLM generates a series of rationales, and those that lead to the reasonable weights. For example in Figure 5, we demonstrate the importance of the CoT method in generating more accurate weights. Without the stepwise reasoning prompt, LLM generates the same weight for both the subject and object visual cues for "looking at", which is clearly unreasonable. However, with the CoT prompt, LLM is able to analyze each cue step by step, leading to a more accurate assignment of weights, *i.e.*, the cues about the subject are relatively more important. In order to standardize the format of the strings generated by LLMs for extracting different components of visual cues and weights, we make certain modifications to the prompts for descriptions and weights[3].

## 3 Experiment

### 3.1 Experiment setup

**Datasets.** We evaluated our method on four zero-shot VRD benchmarks: 1) **VG** [13] contains 26,443 images for testing, each annotated with object and predicate labels to form a scene graph. Following previous works [5], we used the pre-processed VG with 150 object classes. We adopted the 24 semantic predicate classes proposed in [19, 20], as they are more informative and challenging for classifying. 2) **GQA** [14] is a large-scale SGG dataset. We used the same split provided by [21], which contains 8,208 images for testing with 200 object classes. As for predicate classes, we selected

Table 1: Evaluation results on the test set of VG and GQA datasets. † denotes removing the guidance from high-level object category. ⋆ denotes integrated with Filter strategy.

| Data | Method | Predicate Classification | | | | | | | | | | | |
| | | R@20 | △ | R@50 | △ | R@100 | △ | mR@20 | △ | mR@50 | △ | mR@100 | △ |
|---|---|---|---|---|---|---|---|---|---|---|---|---|---|
| VG | CLS | 7.2 | - | 10.9 | - | 13.2 | - | 9.4 | - | 14.0 | - | 17.6 | - |
| | CLSDE | 7.0 | -0.2 | 10.6 | -0.3 | 12.9 | -0.3 | 8.5 | -0.9 | 13.6 | -0.4 | 16.9 | -0.7 |
| | RECODE† | 7.3 | 0.1 | 11.2 | 0.3 | 15.4 | 2.2 | 8.2 | -1.2 | 13.5 | -0.5 | 18.3 | 0.7 |
| | RECODE | **9.7** | 2.5 | **14.9** | 4.0 | **19.3** | 6.1 | **10.2** | 0.8 | **16.4** | 2.4 | **22.7** | 5.1 |
| | RECODE⋆ | **10.6** | 3.4 | **18.3** | 7.4 | **25.0** | 11.8 | **10.7** | 1.3 | **18.7** | 4.7 | **27.8** | 10.2 |
| GQA | CLS | 5.6 | - | 7.7 | - | 9.9 | - | 6.3 | - | 9.5 | - | 12.2 | - |
| | CLSDE | 5.4 | -0.2 | 7.2 | -0.5 | 9.3 | -0.6 | 6.0 | -0.3 | 8.8 | -0.7 | 11.5 | -0.7 |
| | RECODE† | 5.2 | -0.4 | 7.8 | 0.1 | 10.2 | 0.3 | 5.8 | -0.5 | 8.9 | -0.6 | 11.3 | -0.9 |
| | RECODE | **6.3** | 0.7 | **9.4** | 1.7 | **11.8** | 1.9 | **7.8** | 1.5 | **11.9** | 2.4 | **15.1** | 2.9 |
| | RECODE⋆ | **7.0** | 1.4 | **11.1** | 3.4 | **15.4** | 5.5 | **9.4** | 3.1 | **14.8** | 5.3 | **20.4** | 8.2 |

26 semantic predicate classes by referring to VG. 3) **HICO-DET** [15] contains 9,658 testing images annotated with 600 HOI triplets derived from combinations of 117 verb classes and 80 object classes. 4) **V-COCO** [16] comprises 4,946 testing images annotated with 29 action categories.

**Evaluation Metrics.** For SGG datasets (*i.e.*, VG and GQA), we reported **Recall@K (R@K)** which indicates the proportion of ground-truths that appear among the top-K confident predictions, and **mean Recall@K (mR@K)** which averages R@K scores calculated for each category separately [22]. For HOI datasets (*i.e.*, HOI-DET and V-COCO), we reported **mean Average Precision (mAP)** [23].

**Implementation Details.** For the LLM, we employed the GPT-3.5-turbo, a highly performant variant of the GPT model. As for CLIP, we leveraged the OpenAI's publicly accessible resources, specifically opting for the Vision Transformer with a base configuration (ViT-B/32) as default backbone[3].

**Settings.** The bounding box and category of objects were given in all experiments. We compared our RECODE with two baselines: 1) **CLS**, which uses relation-CLasS-based prompts (*e.g.*, "riding") to compute the similarity between the image and text. 2) **CLSDE**, which uses prompts of relation CLasS DEscription as shown in Figure 4(a). Each component of the proposed framework can serve as a plug-and-play module for zero-shot VRD. Specifically, 1) **Filter**, which denotes filtering those unreasonable predictions (*e.g.*, `kid-eating-house`) with the rules generated by GPT[3]. 2) **Cue**, which denotes using description-based prompts (Sec. 2.1). 3) **Spatial**, which denotes using spacial images as additional features. 4) **Weight**, which denotes using dynamic weights generated by GPT to determine the importance of each feature, *i.e.*, visual cue weights.

## 3.2 Results and Analysis

In this work, we evaluated the prediction performance of the proposed framework on two related tasks, *i.e.*, SGG and HOI. The former outputs a list of relation triplet ⟨sub,pred,obj⟩, while the latter just fix the category of sub to human. Overall, our method achieved significant improvement on the two tasks compared to the CLS baseline, which shows the superiority of our method.

**Evaluation on SGG.** From the results in Table 1, we have the following observations: 1) CLSDE showed worse performance than the trivial CLS baseline. This is because the considerable noise in CLSDE which may hinder the model to attend the most distinguishable parts. 2) With the proper guidance, RECODE achieved considerable improvements compared to the baselines, *e.g.*, 0.8% to 6.1% gains on VG and 0.7% to 2.9% gains on GQA. The

Table 2: Evaluation results on the test set of HICO-DET and V-COCO datasets.

| Method | HICO-DET | | | V-COCO | |
| | Full | Rare | Non-Rare | Role 1 | Role 2 |
|---|---|---|---|---|---|
| CLS | 32.3 | 33.2 | 31.8 | 25.5 | 28.6 |
| CLSDE | 32.5 | 33.1 | 32.2 | 25.6 | 28.8 |
| RECODE† | 32.5 | 33.0 | 32.4 | 25.7 | 28.8 |
| RECODE | **32.7** | **33.2** | **32.5** | **26.0** | **29.0** |

performance drops of RECODE† also demonstrated the importance of guidance from high-level object categories during the generation process. 3) Integrated with the filtering strategy, *i.e.*, RECODE⋆, achieved the best performance over all metrics, which suggests that commonsense knowledge is complementary and effective for zero-shot VRD. It also demonstrated that CLIP struggles to distinguish abstract concepts, *i.e.*, relation sensitivity as mentioned in Sec. 1.

Table 3: Ablation studies on different architectures of CLIP. The official released weights are used.

| Architecture | Method | Predicate Classification | | | | | |
|---|---|---|---|---|---|---|---|
| | | R@20 | R@50 | R@100 | mR@20 | mR@50 | mR@100 |
| ViT-L/14 | CLS$^\star$ | 8.3 | 15.0 | 21.5 | 7.6 | 14.2 | 24.2 |
| | RECODE$^\star$ | **11.2** | **19.9** | **28.0** | **9.1** | **18.5** | **28.1** |
| ViT-L/14@336px | CLS$^\star$ | 8.6 | 15.4 | 21.8 | 7.7 | 13.9 | 23.0 |
| | RECODE$^\star$ | **12.1** | **21.1** | **29.2** | **9.7** | **19.5** | **28.2** |
| ViT-B/32 | CLS$^\star$ | 7.5 | 13.7 | 19.4 | 9.1 | 15.9 | 24.0 |
| | RECODE$^\star$ | **10.6** | **18.3** | **25.0** | **10.7** | **18.7** | **27.8** |
| ViT-B/16 | CLS$^\star$ | 8.6 | 15.5 | 22.1 | 9.8 | 17.2 | 25.2 |
| | RECODE$^\star$ | **12.6** | **21.0** | **28.5** | **12.5** | **20.2** | **30.0** |

**Evaluation on HOI.** Since standard evaluation procedure of HOI had filtered out those unreasonable predictions, RECODE$^\star$ was not evaluated here. From the results in Table 2, we can observe that the performance gains were lower than those on SGG, *e.g.*, 0.0% to 0.7% gains on HICO-DET and 0.4% to 0.5% gains on V-COCO. The reasons are two-fold. On the one hand, since the category of subject is always a human, its features are too similar to be distinguished by CLIP. On the other hand, some of the actions are very similar in appearance. For example, distinguishing between actions like "person-throw-sports ball" and "person-catch-sports ball" is challenging due to their visual similarity.

## 3.3 Diagnostic Experiment

**Architectures.** We investigated the impact of changing the architectures of CLIP, as shown in Table 3. From the results, we can observe consistent improvements regardless of the architecture used.

**Key Component Analysis.** The results are summarized in Table 4. The first row refers to the CLS baseline. Four crucial conclusions can be drawn. **First**, with the guidance of Cue, consistent improvements can be observed, *e.g.*, 0.2% to 3.4% gains on R@K w/o Filter and 1.3% to 4.1% gains on R@K with Filter. **Second**, by introducing the spatial feature, the relative position of subject and object is considered, resulting in notable perfor-

Table 4: Analysis of key components on the test set of VG.

| Filter | Cue | Spatial | Weight | Predicate Classification | | | | | |
|---|---|---|---|---|---|---|---|---|---|
| | | | | R@20 | R@50 | R@100 | mR@20 | mR@50 | mR@100 |
| | | | | 7.2 | 10.9 | 13.2 | 9.4 | 14.0 | 17.6 |
| | ✓ | | | 7.4 | 12.3 | 16.6 | 9.0 | 14.0 | 19.5 |
| | ✓ | ✓ | | 9.1 | 13.4 | 17.4 | 9.3 | 15.0 | 20.3 |
| | ✓ | | ✓ | 7.9 | 13.4 | 17.7 | 9.3 | 14.7 | 20.5 |
| | ✓ | ✓ | ✓ | **9.7** | **14.9** | **19.3** | **10.2** | **16.4** | **22.7** |
| ✓ | | | | 7.5 | 13.7 | 19.4 | 9.1 | 15.9 | 24.0 |
| ✓ | ✓ | | | 8.8 | 15.9 | 23.5 | 10.3 | 17.2 | 26.2 |
| ✓ | ✓ | ✓ | | 9.3 | 16.3 | 22.5 | 10.1 | 18.1 | 25.5 |
| ✓ | ✓ | | ✓ | 10.0 | 17.5 | 24.8 | 10.4 | 17.8 | 26.7 |
| ✓ | ✓ | ✓ | ✓ | **10.6** | **18.3** | **25.0** | **10.7** | **18.7** | **27.8** |

mance gains on R@K (0.8% to 1.7%) and mR@K (0.3% to 1.0%) w/o Filter compared to just using Cue. This is because the spatial feature is of importance for relation detection [6]. **Third**, benefiting from the impressive reasoning ability of LLMs, the proposed weighting strategy can determine the importance of different cues, thus achieving further improvements, *e.g.*, 0.5% to 1.1% gains on R@K compared to average aggregation. **Fourth**, by filtering those unreasonable predictions, consistent improvements can be observed. The reason may be that the performance of relation detection of CLIP is not accurate enough. Empirically, commonsense knowledge is a feasible way to filter those noise. Combining all components allows for the best overall performance on all evaluation metrics.

**Case study.** To investigate the most important regions for distinguishing relations, we visualized the attention map given different images and prompts (*cf.*, Figure 6). From the visualization of class-based prompts, we can observe that CLIP may attend those regions unrelated to the query prompts, *e.g.*, focusing on the body of a person given relation "growing on". We attribute this phenomenon to the insufficient information within given prompts, which is also our motivation to introduce visual cue descriptions. As for description-based prompts, CLIP can attend to right regions with the guidance of descriptions, *e.g.*, focusing on colorful patterns on the product given relation "painted on".

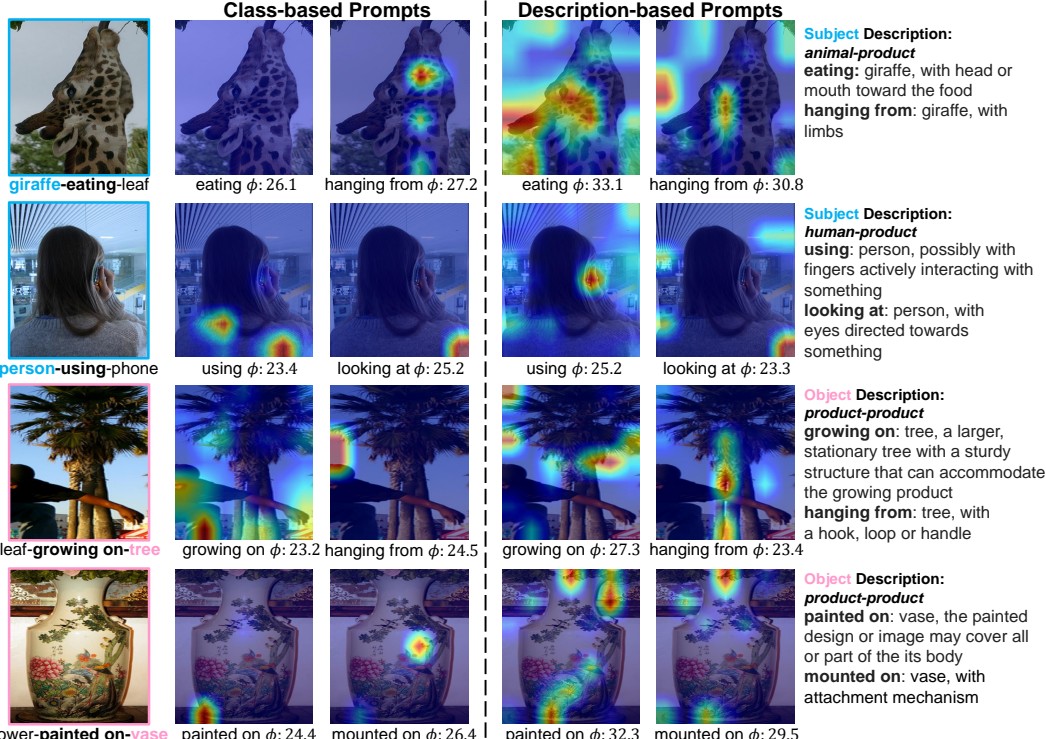

Figure 6: Visualization of CLIP attention maps on input images with different prompts. The right side shows the partial description-prompts generated for each predicate category given the high-level object category. They are used to generate the corresponding attention maps on the right.

# 4 Related Work

**Visual Relation Detection (VRD)** aims to predict the relationships of given subject-object pairs, which can be viewed as a pair-wise classification task and have been widely studied in the image domain, *e.g.*, scene graph generation (SGG) [5, 6, 22, 24] and human-object interaction (HOI) detection [25, 26, 27]. Previous solutions mainly focus on learning representations from the training samples on pre-defined categories, which may suffer from noisy annotations [22] or long-tailed predicate distribution [6, 28] and are far from the needs of the real-world scenarios. Recently, some attempts [8, 29] adopted prompt-tuning [30] to predict unseen categories during inference. However, since the learnable prompts may be overfitting when trained on seen categories, their performance is sensitive to the split of seen/unseen categories [31]. In contrast, our method can predict the relationships directly without any training samples, and has better interpretability and generalization ability, especially in rare informative relation categories.

**Zero-shot Visual Recognition** enables the model to recognize new categories that it has never seen during training, which is one of the research hotspots in the vision community. Aligning visual representations to pre-trained word embeddings (*e.g.*, Word2Vec [32] and GloVe [33]) is an intuitive and feasible way to achieve this goal [34]. More recently, VLMs, which use contrastive learning [35] to learn a joint space for vision and language, have demonstrated their impressive zero-shot ability [1]. Therefore, many zero-shot works [36, 37, 38] adopted such VLMs as their basic component to use the knowledge of the learned joint space. However, most of them only utilized the *class name* of unseen categories during inference, which makes an over-strong assumption that the text encoder project proper embeddings with only category names [11]. Then, Menon and Vondrick [11] proposed to query LLMs for the rich context of additional information. Nonetheless, it is non-trivial to apply such paradigms to VRD as discussed in Sec. 1. To the best of our knowledge, we are the first to leverage both LLMs and VLMs for VRD in an efficient, effective, and explainable way.

## 5  Conclusion

In this paper, we proposed a novel approach for zero-shot Visual Relationship Detection (VRD) that leverages large language models (LLMs) to generate detailed and informative descriptions of visual cues for each relation category. The proposed method addresses the limitations of traditional class-based prompts and enhances the discriminability of similar relation categories by incorporating specific visual cues. Moreover, we introduced a chain-of-thought method that breaks down the problem into smaller, more manageable pieces, allowing the LLM to generate a series of rationales for each visual cue and ultimately leading to reasonable weights. Our experiments on four benchmark datasets demonstrated the effectiveness and interpretability of our method.

**Acknowledgement.** This work was supported by the National Key Research & Development Project of China (2021ZD0110700), the National Natural Science Foundation of China (U19B2043, 61976185), and the Fundamental Research Funds for the Central Universities (226-2023-00048). Long Chen is supported by HKUST Special Support for Young Faculty (F0927), and HKUST Sports Science and Technology Research Grant (SSTRG24EG04).

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
