# Zero-shot Visual Relation Detection via Composite Visual Cues from Large Language Models
## ***********Supplementary Document************

## Appendix

This supplementary document is organized as follows:

- The details about stimulated spatial images generation mentioned in Sec. 2.1 is shown in Sec. A.
- The prompts for the high-level object category generation (*cf.*, Sec. 2.2.1), visual cue description (*cf.*, Sec. 2.2.1), visual weight determination (*cf.*, Sec. 2.2.2), and filtering strategy (*cf.*, Sec. 3.1) are presented in Sec. B.
- The implementation details mentioned in Sec. 3.1 are provided in Sec. C.
- Additional experimental and qualitative results are reported in Sec. D.
- The broader impacts of the proposed method are discussed in Sec. E.
- The limitations of the proposed method are presented in Sec. F.

## A   Stimulated Spatial Images Generation

We propose to simulate the spatial relationship between the subject and object by generating a finite set of spatial images, as mentioned in Sec. 2.1. Each spatial image represents the bboxes of the subject and object, where the subject's bounding box is visually denoted by a red box, and the object's bounding box is denoted by a green box. We define four essential attributes, namely shape, size, relative position, and distance, to describe the spatial relationships between the subject and object. These attributes are calculated based on various characteristics, including the aspect ratio $\rho$ and area $A$ of the bounding boxes, the cosine similarity $sim(\cdot, \cdot)$, and the Euclidean distance $d$ between their centers. By assigning different values to these attributes, we can generate a diverse set of simulated spatial images. Given a specific triplet, we calculate the value of each attribute based on the characteristics of the subject and object. Next, we search for the most suitable spatial image in the simulated set by matching these attribute values. This matching process involves comparing the calculated attributes of the triplet with the corresponding attribute ranges in the simulated set. For instance, the aspect ratios and areas of the subject and object bounding boxes determine their shape and size attributes, while the cosine similarity and Euclidean distance between their centers contribute to the relative position and distance attributes. By utilizing this approach, we can effectively simulate various spatial relationships between the subject and object to improve computing efficiency. The detailed procedures of the algorithm are provided in Algorithm 1.

37th Conference on Neural Information Processing Systems (NeurIPS 2023).

---

**Algorithm 1:** Spatial Relationship Simulation

---

**Require:** Bounding boxes $b_s$ and $b_o$ for the subject and object of the triplet.

**Ensure:** Spatial image $I_t$ corresponding to the triplet.

1: **Step 1: Generate simulated spatial image set**
2: Define the attributes: shape, size, relative position, and distance.
3: Specify the corresponding value intervals for each attribute.

- Shape: horizontal, vertical, and square denoted as $\{\mathcal{H}, \mathcal{V}, \mathcal{Q}\}$.
- Size: small, medium, and large denoted as $\{\mathcal{S}, \mathcal{M}, \mathcal{L}\}$.
- Relative position: above ($\uparrow$), below ($\downarrow$), left ($\leftarrow$), right ($\rightarrow$), top-left ($\nwarrow$), top-right ($\nearrow$), bottom-left ($\swarrow$), and bottom-right ($\searrow$).
- Distance: small, medium, large denoted as $\{\mathcal{S}, \mathcal{M}, \mathcal{L}\}$.

4: Draw spatial images by combining all attribute values to create a set of simulated spatial images denoted as $\{I_i\}$.
5: **Step 2: Calculate attributes of the given triplet**
6: Calculate the centers $c_s$ and $c_o$ of the subject and object bounding boxes, respectively.
7: Calculate the aspect ratios $\rho_s = \frac{\text{width}(b_s)}{\text{height}(b_s)}$ and $\rho_o = \frac{\text{width}(b_o)}{\text{height}(b_o)}$ of the subject and object bounding boxes, respectively.
8: Calculate the areas $A_s = \text{width}(b_s) \times \text{height}(b_s)$ and $A_o = \text{width}(b_o) \times \text{height}(b_o)$ of the subject and object bounding boxes, respectively.
9: Calculate the cosine similarity $sim(c_s, c_o) = \frac{c_s \cdot c_o}{\|c_s\| \cdot \|c_o\|}$ and the Euclidean distance $d_{s,o} = \sqrt{(c_s - c_o)^2}$ between the centers $c_s$ and $c_o$.
10: **Step 3: Match a spatial image in the simulated set**
11: Find the shape and size intervals $S_s$, $S_o$, $L_s$, and $L_o$ that $\rho_s$, $\rho_o$, $A_s$, and $A_o$ belong to.
12: Find the relative position and distance interval that $sim(c_s, c_o)$ and $d_{s,o}$ belong to.
13: Match the spatial image $I_t$ in $\{I_i\}$ by combining the appropriate attributes.
14: **return** $I_t$

---

# B  Prompts

In this section, we present prompts for high-level object category generation (*cf.*, Sec. 2.2.1), visual cue description (*cf.*, Sec. 2.2.1), visual weight determination (*cf.*, Sec. 2.2.2), and unreasonable predicate filtering (*cf.*, Sec. 3.1).

**High-level Object Class Generation Prompt.** To facilitate the classification of low-level object categories into high-level object categories, we provide the following prompt:

```
Given the low-level object categories: [ALL OBJ CLS].
Please classify each low-level object category into high-level object
categories ["human", "animal", "product"] based on their most common
semantics in visual relation detection.
Ensure that body parts and similar categories are not classified as "human".
Note that human beings engaged in certain activities must be classified as
"human"!
```

In this prompt, we provide a list of low-level object categories: [ALL OBJ CLS] that need to be categorized into high-level object categories(*i.e.*, "human", "animal", and "product"). The prompt instructs the LLMs to assign the low-level object categories to the most appropriate high-level object category based on their prevalent semantics in visual relation detection. This prompt guides the model to understand the distinctive characteristics and visual cues associated with different object categories, contributing to accurate descriptions of visual cue for each relation category.

**Visual Cue Description Prompt.** We present guided relation component description prompt for generating the descriptions of visual cue, specifically designed for the relation class "REL CLS" when provided with the High-Level (HL) categories of the subject and object, *i.e.*, "SUB HL CLS" and "OBJ HL CLS". The prompt is structured as follows:

```
Known: a visual triplet is formulated as [subject, predicate, object].
Note that:
```

```
  [position] must not include nouns other than subject and object!
  [position] must contain
    [orientation: ("above", "below", "left", "right", "inside"),
     shape: ("horizontal", "vertical", "square"),
     distance: ("small distance", "mid distance", "large distance")]!

Describe the visual features of the predicate "sitting on" in a photo,
when subject belongs to [human], object belongs to [product]:
  [subject]:
    - with legs.
    - with hip.
  [object]:
    - with flat surface.
  [position]:
    - square subject above horizontal object with a small distance.

Describe the visual features of the predicate "REL CLS" in a photo,
when subject belongs to [SUB HL CLS], object belongs to [OBJ HL CLS]:
```

The prompt is divided into four distinct parts: setting, constraint, example, and question. **Setting:** The setting (*i.e.*, "`Known...`") provides specific roles and known conditions for the LLMs to operate within. **Constraint:** The constraint (*i.e.*, "`Note that...`") outlines some limitations or constraints on the output generated by the LLMs. **Example:** The example (*i.e.*, the example of "sitting on") serves as a guide for the model to produce similar output in an in-context learning [1, 2] manner, which is also generated by LLM. **Question:** Finally, the question (*i.e.*, "`Describe...`") prompts the model to generate a description of the visual features that are specific to the relation being considered. This comprehensive prompt structure aids in more reasonable and standardized generation of visual cue descriptions for subject, object, and spatial components for each relation category.

**Visual Cue Weight Prompt.** The prompt is designed to determine the visual cue weights for subject (SUB CUES), object (OBJ CUES), and spatial (POS CUES) in relation classification. It is structured as follows:

```
Suppose you are a relation classification model.

Given: subject belongs to [human] and object belongs to [product].
The visual features of subject:
  ["with eyes directed towards the object", "with head upright"].
The visual features of object:
  ["with visible features such as front, display, or screen"].
The visual features of position:
  ["subject positioned either above, below, left or right of the object at
   a mid distance"].
Q: How do you weight these visual features (subject, object, position) to
determine the predicate is "looking at"? The sum of weights must be 1.0!
A: Let's think step by step!
First, we need to consider the importance of the subject's visual features.
Since the direction of the eyes and head position strongly indicate the
focus of attention, we will give them a weight of 0.6. Next, we need to
consider the importance of the object's visual features. Since the visible
features such as front, display, or screen indicate that the object is
something that can be looked at, we will give them a weight of 0.3. Finally,
we need to consider the importance of the position visual features. Since
the relative position of the subject and object at a mid-distance helps us
understand that the subjects are looking at the object in question, we will
give them a weight of 0.1.
Therefore, we can weight these visual features as follows:
Weight("looking at") = 0.6 * Weight(visual features of subject)
                     + 0.3 * Weight(visual features of object)
                     + 0.1 * Weight(visual features of position).
```

```
Given: subject belongs to [SUB HL CLS] and object belongs to [OBJ HL CLS].
The visual features of subject:
  [SUB CUES].
The visual features of object:
  [OBJ CUES].
The visual features of position:
  [POS CUES].
Q: How do you weight these visual features (subject, object, position) to
determine the predicate is "REL CLS"? The sum of weights must be 1.0!
A: Let's think step by step!
```

The prompt is also divided into four distinct parts: setting, constraint, example, question. **Setting:** The setting (*i.e.*, "Suppose...") establishes the role and perspective of the model in the task. **Constraint:** The constraint (*i.e.*, "The sum of weights must be 1.0!") provides some limitations or constraints on the output generated by the LLMs. **Example:** The example (*i.e.*, the example of determining the predicate "looking at") serves as a guide for the LLMs to understand the context and expected output. **Question:** The question (*i.e.*, "How do...") prompts the model to determine the weights assigned to visual cues in order to classify the given predicate. Additionally, the stepwise prompt "Let's think step by step!" guides the LLMs to incrementally analyze the problem and generate rationales, which lead to more reasonable determination of visual cue weights.

**Filter Prompt.** The prompt is used to filter unreasonable sub-pred and obj-pred categories. The prompt for sub-pred is as follows:

```
Q: Can the window be sitting on something?
After thinking about it, just answer "Yes" or "No"!
A: Let's think step by step!
It is possible for a window to be sitting in something, such as a frame
or sill.
Answer is Yes.

Q: Can the {SUB CLS} be {REL CLS} something?
After thinking about it, just answer "Yes" or "No"!
A: Let's think step by step!
```

The prompt also adopts an in-context learning approach, leveraging illustrative examples to enhance comprehension of the situation. The stepwise prompt stimulates logical reasoning, facilitating the LLMs in rendering more robust judgments by leveraging the provided information.

## C  Implementation Details

Our RECODE does not require a training process and can be directly tested on a NVIDIA 2080 Ti GPU. We pre-computed the visual features encoded by CLIP for each bounding box, enabling us to set the batch size to 512. For the LLM, we utilized the GPT-3.5-turbo, a highly performant variant of the GPT model. As for CLIP, we leveraged the OpenAI's publicly accessible resources, specifically opting for the Vision Transformer with a base configuration (ViT-B/32) as the default backbone.

## D  Further Analysis

### D.1  Comparison with Training-based Methods

In this section, we compared the proposed **training-free** RECODE framework with those well-designed training-based ones in Table 5. Note that such comparisons are unfair as training-based frameworks can learn the underline patterns and data distribution from the training set. For completeness, we still reported the results and investigate the performance gap between training-based frameworks and RECODE. Specifically, we compared the proposed RECODE with several relevant baselines, including triplet-level zero-shot VRD [3, 4], few-shot VRD [5], and category-level zero-

Table 5: Comparison with SOTA VRD methods on the VG dataset. Note that none of these methods can be applied in the **training-free** zero-shot setting.

| Model | No Training | Unseen Relation | Training Data Source | Predicate Classification | | |
|---|---|---|---|---|---|---|
| | | | | zR@20 | zR@50 | zR@100 |
| Motifs [3] | ✗ | ✗ | VG | 8.9 | 15.2 | 18.5 |
| COACHER [4] | ✗ | ✗ | VG& ConceptNet | 28.2 | 34.1 | 37.2 |
| DPL [5] | ✗ | ✗ | VG | 6.0 | 7.7 | 9.3 |
| CaCao [6] | ✗ | ✓ | VG&CC3M&COCO | 17.2 | 21.3 | 23.1 |
| RECODE | ✓ | ✓ | - | 8.2 | 16.1 | 23.2 |

shot VRD [6]. Since all of them can not detect relations without training, we reported Zero-shot Recall@K (**zR@K**), which only calculates the Recall@K for those unseen **triplet** categories.

- Triplet-level zero-shot VRD methods. Motifs [3] is a traditional strong baseline without explicitly modeling the nature of zero-shot. COACHER [4] explicitly models the nature of zero-shot, and takes the power of the common sense from ConceptNet resulting in better performance.
- Few-shot VRD methods. DPL [5] is a few-shot baseline, which mainly investigates making predictions with a few examples (here we evaluate 1-shot).
- Category-level zero-shot VRD methods. CaCao [6] also explicitly models the nature of zero-shot, and leverages language information from captions of CC3M and COCO for enhanced performance.

Surprisingly, even without training, RECODE still achieves competitive results, with zR@20, zR@50, and zR@100 of 8.2%, 16.1%, and 23.2%, respectively. This signifies its potential in handling unseen categories, due to the effective visual cues and inference mechanisms.

## D.2 Ablation on Different Class-based Prompts

Table 6: Ablation studies of different class-based prompts on the test set of VG

| Class-based Prompt | Method | Predicate Classification | | | | | |
|---|---|---|---|---|---|---|---|
| | | R@20 | R@50 | R@100 | mR@20 | mR@50 | mR@100 |
| [REL CLS]-ing/ed | CLS$^\star$ | 7.5 | 13.7 | 19.4 | 9.1 | 15.9 | 24.0 |
| | RECODE$^\star$ | **10.6** | **18.3** | **25.0** | **10.7** | **18.7** | **27.8** |
| a photo of [REL CLS] | CLS$^\star$ | 11.7 | 19.2 | 26.2 | 10.9 | 19.4 | 27.1 |
| | RECODE$^\star$ | **13.5** | **21.8** | **28.8** | **12.2** | **19.6** | **28.4** |

We conducted an ablation study to investigate the impact of different class-based prompts on zero-shot VRD performance. The class-based prompts were manually designed to generate text embedding for relation classification. We compared two types of class-based prompts: 1) "[REL-CLS]-ing/ed" prompt, where [REL-CLS] represents the name of relation category. For example, for the relation class "milk", the prompt would be "milking". 2) "a photo of [REL-CLS]" prompt. For example, for the relation class "riding", the prompt would be "a photo of riding".

Table 6 summarized the results. Our method achieved improved scores across various metrics for both types of prompts. With the "[REL-CLS]-ing/ed" prompt, we observed significant gains (3.1% to 5.6%) on R@K. Similarly, when using the "a photo of [REL-CLS]" prompt, we achieved the highest R@100 score of 28.8% and mR@100 of 28.4%. These results indicated that our method consistently outperforms the CLS baseline, regardless of the specific prompt type used. The effectiveness of our method suggested a promising solution for zero-shot VRD tasks.

## D.3 Interpretability Analysis

To gain a deeper understanding of the interpretability of our RECODE in VRD, we conducted an in-depth analysis comparing its predictions with CLS baseline that utilizes only class-based prompts.

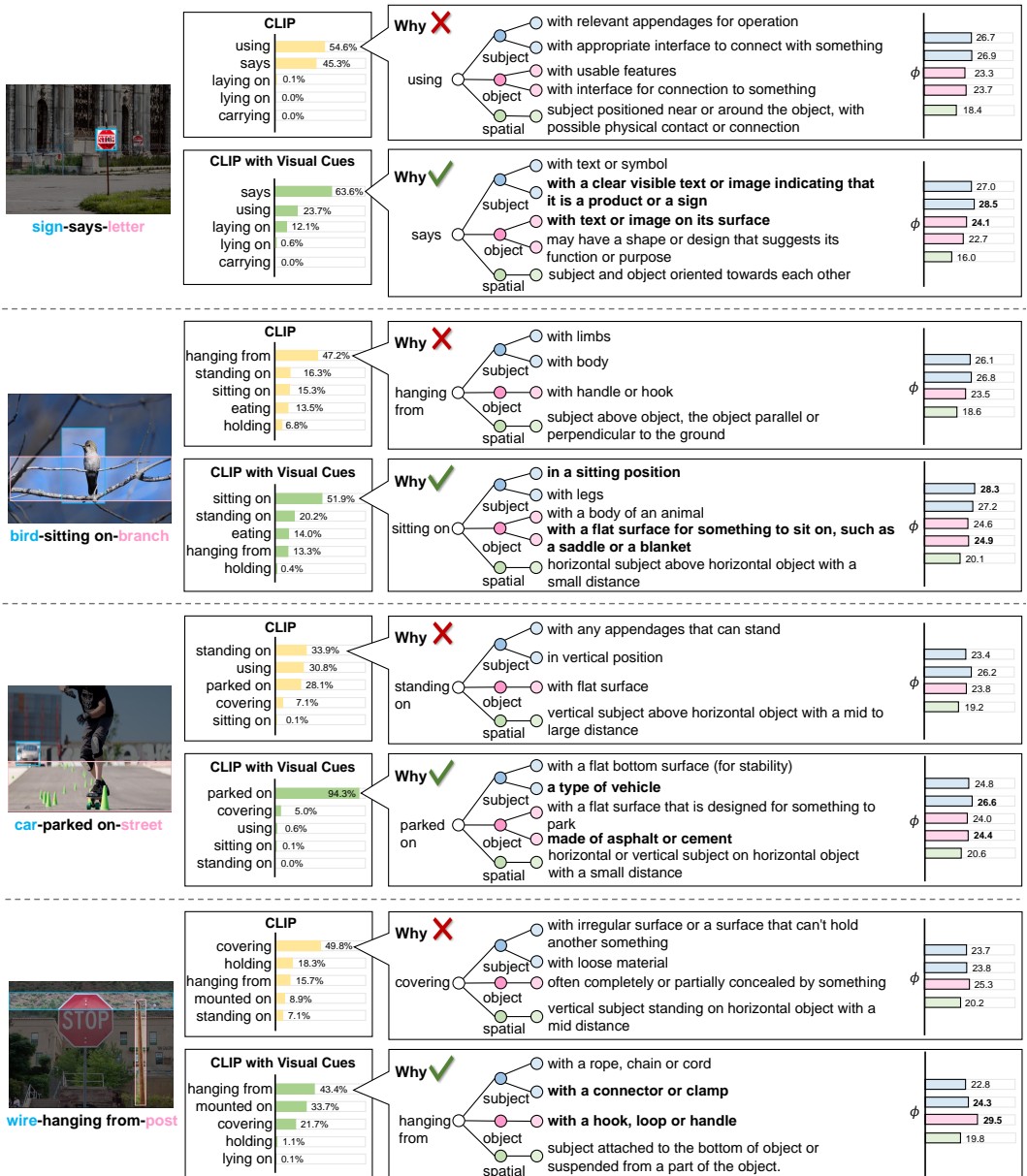

Figure 7: A comparative analysis of predictions made by RECODE and a baseline with class-based prompts on the test set of VG. It illustrates how our method offers interpretability to the VRD through the similarity $\phi$ between the image and the description-based prompts.

By evaluating the similarity between the description-based prompts and the corresponding visual features, we revealed the underlying reasons for the accuracy of RECODE's predictions and the inaccuracies of the CLIP baseline.

Figure 7 presented qualitative comparisons of RECODE and the CLS baseline on challenging examples from the VG dataset. Our description-based prompts significantly improve CLIP's understanding of the various relation categories, leading to more accurate predictions. Taking the top image of Figure 7 as an example, RECODE accurately predicts the "says" relation category by identifying the presence of visual features associated with "text or image". In contrast, the failure case of the "using" relationship category predicted by the CLS baseline can be attributed to the absence of distinctive visual features related to "usable feature", as highlighted by our description-based prompts.

# E Broader Impacts

Like every coin has two sides, using our method will have both positive and negative impacts.

**Positive Impacts. Firstly**, RECODE emphasizes the importance of pairwise recognition, encouraging researchers to develop more diverse and comprehensive recognition models. By focusing on the relationships between object pairs, we inspire the exploration of a broader range of relationship types and promote a deeper understanding of complex interactions between objects (*e.g.*, n-tuple interaction), especially in the zero-shot setting without any extra training stage. **Secondly**, our method introduces the incorporation of spatial information in visual relation detection. By considering spatial cues and relationships between objects, we highlight the significance of spatial information in understanding object interactions. This not only improves the accuracy of relation detection but also encourages researchers to explore the integration of other useful auxiliary information. This can include incorporating contextual information, temporal relationships, or other relevant cues that can enhance recognition performance. **Thirdly**, our method promotes the use of Chain-of-Thought prompting with LLMs for weight assignment in recognition tasks. By leveraging the knowledge and capabilities of LLMs, we enable the generation of more informed and reasonable weights for different components of the recognition process. This improves the interpretability of the recognition results and opens up new possibilities for utilizing the vast knowledge and capabilities of language models to enhance recognition systems.

**Negative Impacts.** However, we also acknowledge that there are potential negative impacts associated with the use of our method. For example, the reliance on LLMs could lead to the perpetuation of biases and inequalities present in the data used to pre-train these models.

In conclusion, the proposed method for zero-shot visual relation detection brings about positive impacts by inspiring more complex recognition models under the zero-shot setting, highlighting the significance of contextual cues, and promoting the use of LLMs for weight assignment. It is essential to continue exploring ways to address potential negative impacts and ensure the responsible and ethical use of these advancements in our community.

# F Limitations

As the first zero-shot visual relation detection work using LLMs, our method still has some limitations: **1) Firstly**, we did not specifically evaluate spatial relation categories (*e.g.*, "on", "under") and ownership relation categories (*e.g.*, "belong to"). In this work, our method mainly focuses on classifying semantic predicate groups based on visual cue descriptions. However, by extensive empirical results, these spatial and ownership relationships can be easily predicted from only spatial positions or object categories. **2) Secondly**, our framework assumes the availability of ground truth bounding boxes and object categories for relation classification. However, in real-world scenarios, object detection can introduce errors or uncertainties. **3) Thirdly**, to avoid overmuch queries to LLMs, our approach proposes a trade-off solution and only relies on coarse-grained triplet category descriptions. However, this simplification may not capture fine-grained nuances in different visual relationships. Using more detailed and comprehensive descriptions (with more LLM queries) could potentially further improve the performance. **4) Fourthly**, the accuracy and correctness of the visual cue descriptions are not guaranteed. Despite efforts to ensure quality, errors or incomplete information may be present. It is essential to validate and verify cue descriptions for reliable results.