# OpenReview forum: "Zero-shot Visual Relation Detection via Composite Visual Cues from Large Language Models"
_NeurIPS.cc/2023/Conference — NeurIPS 2023 poster_

### Official Review · Reviewer_EWRp · 2023-06-30

**Soundness:** 3 good
**Presentation:** 3 good
**Contribution:** 2 fair
**Rating:** 5
**Confidence:** 4

**Summary:**

In this paper, the authors developed a joint model of CLIP and LLM to solve the task of Visual relation detection. In this model, images are encoded into a triplet,~\ie, object, subject and spatial branches. Then it leverages large language models (LLMs) to generate description-based prompts (or visual cues) for each component. Experiments on four VRD benchmarks shows good results compared to the baseline models.

**Strengths:**

+ Interesting and good applications of LLMs including GPT-3.5 and large multi-modality pertaining models~\eg, CLIP.
+ Show higher performance than baseline methods.
+ Good storytelling to readers understand the key idea.

**Weaknesses:**

### Technical Novelty and main ideas
1. This paper pays its major attention to designing and using LLM~\ie, GPT-3.5 to facilitate the deduction of multi-modal models. Many contributions lie in the design and feeding prompts into LLMs. This contribution seems insignificant to me and seems not generalizable for future LLM using different architectures with GPT-3.5. Besides, the authors also lack deep investigation into the improvements of the Chain of Thought (CoT).  Although the overall application and using LLMs seem interesting. but the solid contributions of this paper are not clear to me.

### Presentation and motivation issues
The descriptions of CoT and experimental results are somehow unclear.  Besides, the novelty of using CoT in this task is not sufficient. Designing prompts

2. The reviewers tested the same prompt using GPT-3.5, while in this case, with or without Chain of Thought (CoT) does not show significant differences. (with CoT: avg: 0.51, 0.3, 0.19 for s,o,p; w/o CoT: avg: 0.59, 0.33, 0.14 for s,o,p) The reviewer understands the results may not be stable but is still unclear about this case.

3. Why did Figure 5(a) choose 0.4, 0.4, 0.2? The authors should explain how `obviously unreasonable` on lines 188 - 190 is `obviously unreasonable`.

### Experimental issues

4. For Figure 2 and Figure 7 in the supplementary material.
a) Where did the cues for the CLS baseline displayed by the sentence come from? Line 214 of the text states to use "relational CLasS-based hints (e.g., ride)", which are somehow inconsistent with the statements. Can the authors explain in more detail how the CLS baseline is tested? Where does the performance difference come from if two settings use the same prompts?
5. Although the newly proposed setting, the authors do not discuss other similar works.  PEVL[1] and STIP[2]. Differences and relation discussion could help.

[1] Yao, Y., Chen, Q., Zhang, A., Ji, W., Liu, Z., Chua, T. S., & Sun, M. (2022). PEVL: Position-enhanced pre-training and prompt tuning for vision-language models. _arXiv preprint arXiv:2205.11169_.

[2] Zhang, Y., Pan, Y., Yao, T., Huang, R., Mei, T., & Chen, C. W. (2022). Exploring structure-aware transformer over interaction proposals for human-object interaction detection. In _Proceedings of the IEEE/CVF Conference on Computer Vision and Pattern Recognition_ (pp. 19548-19557).

6. The time efficiency of different compared methods in Tab.4 should also be provided.
7. [Experimental Description] For the conjecture in line 247, does deduction give evidence for a confusion matrix? The reviewer did a cursory check of the dataset and there are a total of 117871 annotations, These provided actions seem to be rare.

**Questions:**

Please refer to the weakness section.

Besides the technical novelty, this manuscript exist many unclear implementation problems and insufficient experimental results, especially a lack of discussion with other works using LLMs.

 I hope the authors could solve these concerns to make this paper more readable.

**Limitations:**

The authors clearly discussed the limitations of the proposed method. These discussions are fair.

---

> ### Author Rebuttal · Authors · 2023-08-09
>
> Thank you for the detailed comments. We are willing to address all your questions.
> ## Technical Novelty and Main Ideas
> We appreciate your attention to prompt design for GPT-3.5. While the proposed prompts are tailored to GPT-3.5, the core idea of using compositional visual cues for VRD and utilizing prompts for weight allocation is a promising approach that can be extended to future LLMs with various architectures beyond GPT-3.5. Further elaboration on these contributions is available in the **global response**.
> ## Presentation and Motivation Issues
> In our study, we emphasize the varying importance of different components. Our main aim is to have LLMs assess the importance of subject, object, and spatial components. Reviewing examples from the reviewer, it's evident that the subject holds more importance than the object. To further validate this, we gathered statistics about the "look at" relation samples in VG dataset: 28 subject categories and 95 object categories. Such concentration on subjects implies their greater role, making the weights (0.4, 0.4, 0.2) for subject, object, and spatial components (equal weight for subject and object) appear less reasonable. We'll employ milder terms in place of "obviously unreasonable''.
>
> Although the results of weight assignment without CoT may in some cases be similar to those with CoT, the motivation for integrating CoT is to enable LLMs to "think before they act". This requires LLMs to ask themselves: "Why am I weighting this way?", which in most cases leads to reasonable weights. As suggested in a concurrent work [1], GPT may be focusing on the last sentence "The sum of weights must be 1.0!", ignoring other important context. We also investigated the effect of introducing CoT, as shown in Table_R 4. The experimental results demonstrated that introducing CoT achieves consistent improvements, which solidly proves its effectiveness.
>
> **Table_R 4**: Comparison with or without CoT on the VG dataset.
> |CoT|R@20|R@50|R@100|mR@20|mR@50|mR@100|
> |-----|------|------|-------|-------|-------|--------|
> |❌|9.5|17.3|24.6|10.2|18.0|25.6|
> |✔️|10.6|18.3|25.0|10.7|18.7|27.8|
> ## Experimental Issues
> - **Setting.** 1) *Cues for the CLS.* We'd like to clarify that the CLS baseline does not involve the utilization of visual cues (description-based prompts). 2) *CLS baseline.* In a manner similar to various previous object classification approaches, CLS employs relational-class-based prompts as discussed in Line 102. For instance, it adopts prompts like "[REL-CLS]-ing/ed", as illustrated by the example "riding" mentioned in Line 214. CLS obtains the classification score by calculating the similarity between the text feature of class-based prompt of each relation category and the visual feature. 3) *Difference.* In Figure 2 and Figure 7, the baseline "CLIP" (CLIP) relies solely on class-based prompts for generating predictions. While the method "CLIP with Visual Cues" (RECODE) takes advantage of both class-based prompts and description-based prompts.
>
> - **Discuss Similar Works.** Thank you for pointing out the relevance of discussing other similar works and providing references to PEVL, STIP, and other LLM work.
>   - *PEVL.* It focuses on proposing a new pre-training approach for vision-language models, positioning itself as a competitor to CLIP. On the other hand, our work is centered around training-free zero-shot settings, which require no training data and enable direct predictions without any fine-tuning.
>   - *STIP.* It emphasizes traditional fully-supervised methods for human-object interaction (HOI) detection, requiring training and evaluating on the same category set. In contrast, our approach focuses on training-free zero-shot settings, which do not rely on any training data.
>   - *Other LLM work.* The primary application of LLMs has predominantly focused on object classification. Directly extending these methods to visual relation detection (VRD) remains limited (cf, Line 301-302). Our work distinguishes itself as the pioneering effort to explore LLMs in the VRD domain, harnessing both LLMs and vision-language models (VLMs) to address VRD tasks efficiently, effectively, and with interpretability.
>
>   In the revised manuscript, we will include this comprehensive discussion.
>
> - **Time Efficiency.**
> We investigated the time efficiency of each component in RECODE in Table_R 5. Specifically, we calculate the time required to infer each triplet and take the average. Regarding visual cues, their impact on latency remains marginal, amounting to a mere 14.5ms. In terms of spatial features, the computation of similarity demands 13ms. Notably, the spatial similarity in RECODE can be precomputed, given the spatial image within a finite set. Consequently, the calculation of spatial similarity doesn't lead to an increase in inference time, as it's retrieved from the precomputed list. It is pertinent to highlight that the weighting strategy inherent in RECODE does not encompass feature extraction, thereby resulting in a latency close to 0.
>
> **Table_R 5**: Analysis of key components on the VG dataset. Time (ms) represents the computation time of each triplet. The (•) represents the time when spatial component is retrieved offline.
> |Cue|Spatial|Weight|R@20|R@50|R@100|mR@20|mR@50|mR@100|Time(ms)|
> |---------------|---------|--------|------|------|-------|-------|-------|--------|-----------|
> ||||7.2|10.9|13.2|9.4|14.0|17.6|46.7|
> |✔️|||7.4|12.3|16.6|9.0|14.0|19.5|61.2|
> |✔️|✔️||9.1|13.4|17.4|9.3|15.0|20.3|74.2(61.2)|
> |✔️||✔️|7.9|13.4|17.7|9.3|14.7|20.5|61.2|
> |✔️|✔️|✔️|**9.7**|**14.9**|**19.3**|**10.2**|**16.4**|**22.7**|74.2(61.2)|
>
> - **Confusion Matrix.**
> The conjecture in Line 247 is verified in the confusion matrix. As shown in Fig\._R4 (cf, **PDF file**), we found various similar appearances which may hinder the detection, e.g., "eat at" vs "sit at", "ride" vs "straddle", etc.
>
> [1] Lost in the middle: How language models use long contexts. arXiv preprint arXiv:2307.03172, 2023.

---

> ### Author Response · Authors · 2023-08-18
>
> Thank you for your careful and thorough review once again. The deadline for our discussion is approaching. If you have any further concerns or questions, welcome to discuss with us.

---

> > ### Comment · Reviewer_EWRp · 2023-08-19
> > **Replying to rebuttal of authors**
> >
> > Thanks for the detailed response.
> >
> > The response addressed several of my questions but did not alleviate my concerns about the weight assignment. The authors gave an intuitive explanation and experiments for their CoT, however, there is a huge gap before the experimental evidence can support the explanation. The results can only show that the weights selected by the authors are effective, but cannot prove that in general, the weights assigned by the proposed CoT have a significant advantage compared to baseline and the used weights are not cherry-picked by humans. The same gap also shows in the selections of visual cues, as Reviewer CZ8p mentioned. As there are many unstated details between the GPT's output and the actual cues used, I would like to keep my original score.

---

> > > ### Author Response · Authors · 2023-08-19
> > > **Response (Part one)**
> > >
> > > Thanks again for your careful review, but we still want to clarify the following points:
> > > - To reiterate our contribution, we emphasize that **our insight lies in recognizing the distinct significance of subjects, objects, and spatial visual cues**. A simple mean-based approach leads to suboptimal performance. Thus, we adopt GPT to assess the importance of all components with respect to each category.
> > >  -  Besides, if you have used the **web platform of ChatGPT**, you may not get the same results as we do. This is because the web platform keeps all the context (**the next output may refer to the previous input and output**), whereas our method clears the context on each query by using **API**.
> > > - Moreover, several papers have shown that CoT can produce more reasonable results. Inspired by concurrent works in LLMs, we introduce context learning to standardize the output format (examples are given "painted on" in the code). In practice, GPT tends to generate the same weights as in the given example (0.4, 0.4 ,0.2), if we do NOT introduce CoT.  We use a "for" loop to iteratively generate "looking at" weights using the two provided code segments (cf, **code for generating weights in Response Part Two**). The resulting ten consecutive output sets serve as concrete evidence to prove that CoT is more reasonable for our experiment. In order not to violate the review principle, we will return the results of running jupyter to AC in the form of anonymous link to ensure the authenticity of the results in the following Tables.
> > >
> > > **Table: With CoT**
> > >
> > > | Run Times | 1     | 2     | 3     | 4     | 5     | 6     | 7     | 8     | 9     | 10    |
> > > |----------|-------|-------|-------|-------|-------|-------|-------|-------|-------|-------|
> > > | sub weght | 0.5   | 0.6   | 0.5   | 0.5   | 0.6   | 0.4   | 0.5   | 0.5   | 0.5   | 0.6   |
> > > | obj weght | 0.3   | 0.3   | 0.3   | 0.3   | 0.3   | 0.4   | 0.3   | 0.3   | 0.3   | 0.3   |
> > > | pos weght | 0.2   | 0.1   | 0.2   | 0.2   | 0.1   | 0.2   | 0.2   | 0.2   | 0.2   | 0.1   |
> > >
> > > **Table: Without CoT**
> > >
> > > | Run Times | 1     | 2     | 3     | 4     | 5     | 6     | 7     | 8     | 9     | 10    |
> > > |----------|-------|-------|-------|-------|-------|-------|-------|-------|-------|-------|
> > > | sub weght | 0.4   | 0.4   | 0.4   | 0.4   | 0.5   | 0.4   | 0.4   | 0.4   | 0.5   | 0.3   |
> > > | obj weght | 0.4   | 0.4   | 0.4   | 0.4   | 0.3   | 0.4   | 0.4   | 0.4   | 0.3   | 0.4   |
> > > | pos weght | 0.2   | 0.2   | 0.2   | 0.2   | 0.2   | 0.2   | 0.2   | 0.2   | 0.2   | 0.3   |
> > > -  In addition, we provide the code for generating visual cues at **code for generating visual cues in Response Part Three**.
> > > - Reviewer CZ8p has affirmed that we have addressed the majority of the concerns.
> > >
> > > If you are unclear about any of the steps in the implementation details, please don't hesitate to discuss with us. We are more than willing to address any questions you may have.

---

> > > ### Author Response · Authors · 2023-08-19
> > > **Response (Part two)**
> > >
> > > **Code of Weight Generation with CoT**
> > > ```python
> > > import openai
> > > import json
> > > import time
> > > import random
> > > openai.api_key = "YOUR KEY"
> > > sub_feats = ["with eyes directed towards the object, with head upright"]
> > > obj_feats = ["with visible features such as front, display, or screen"]
> > > pos_feats = ["subject positioned either above, below, left or right of the object at a mid distance"]
> > > prompts = '''
> > > Suppose you are a visual relation(predicate) classification model.
> > > Given: subject belongs to [product] and object belongs to [product].
> > > The visual features of subject: ['with a flat surface', 'with colors or designs'].
> > > The visual features of object: ['the painted design or image may cover all or part of the its body'].
> > > The visual features of position: ['subject is placed on the surface of the object'].
> > > Q: How do you weight these visual features(subject, object, position) to determine the predicate is "painted on"? The sum of weights must be 1.0!
> > > A: Let's think step by step!
> > > First, we need to determine which visual feature is the most important for identifying "painted on" as the predicate. From the given visual features, it seems like the presence of the painted design or image on the object may be the most significant indicators of "painted on". However, the fact that the subject is placed on the surface of the object is also important.
> > > Based on this assessment, we can assign weights to each visual feature as follows:
> > > Weight("painted on") = 0.4 * Weight(visual features of subject) + 0.4 * Weight(visual features of object) + 0.2 * Weight(visual features of position).
> > > Given: subject belongs to [{}] and object belongs to [{}].
> > > The visual features of subject: {}.
> > > The visual features of object: {}.
> > > The visual features of position: {}.
> > > Q: How do you weight these visual features(subject, object, position) to determine the predicate is "{}? The sum of weights must be 1.0!
> > > A: Let's think step by step!
> > > '''.format('animal', 'product', sub_feats, obj_feats, pos_feats,'looking at')
> > > messages=[
> > >     {"role": "user", "content": prompts}
> > > ]
> > > try:
> > >     rsp = openai.ChatCompletion.create(
> > >         model="gpt-3.5-turbo",
> > >         messages=messages,
> > >         timeout=10,
> > >         request_timeout=30,
> > >     )
> > >     rsp = json.loads(json.dumps(rsp))
> > >     content = rsp['choices'][0]['message']['content']
> > >     rel_text = content
> > >     print(rel_text)
> > > except Exception as e:
> > >     print(e.args)
> > >     time.sleep(60)
> > > ```
> > >   **Code of Weight Generation without CoT**
> > >
> > >  ```python
> > > sub_feats = ["with eyes directed towards the object, with head upright"]
> > > obj_feats = ["with visible features such as front, display, or screen"]
> > > pos_feats = ["subject positioned either above, below, left or right of the object at a mid distance"]
> > > prompts = '''
> > > Suppose you are a visual relation(predicate) classification model.
> > > Given: subject belongs to [product] and object belongs to [product].
> > > The visual features of subject: ['with a flat surface', 'with colors or designs'].
> > > The visual features of object: ['the painted design or image may cover all or part of the its body'].
> > > The visual features of position: ['subject is placed on the surface of the object'].
> > > Q: How do you weight these visual features(subject, object, position) to determine the predicate is "painted on"? The sum of weights must be 1.0!
> > > A:
> > > Weight("painted on") = 0.4 * Weight(visual features of subject) + 0.4 * Weight(visual features of object) + 0.2 * Weight(visual features of position).
> > > Given: subject belongs to [{}] and object belongs to [{}].
> > > The visual features of subject: {}.
> > > The visual features of object: {}.
> > > The visual features of position: {}.
> > > Q: How do you weight these visual features(subject, object, position) to determine the predicate is "{}? The sum of weights must be 1.0!
> > > '''.format('animal', 'product', sub_feats, obj_feats, pos_feats,'looking at')
> > > messages=[
> > >    {"role": "user", "content": prompts}
> > > ]
> > > try:
> > > 	rsp = openai.ChatCompletion.create(
> > > 	   model="gpt-3.5-turbo",
> > > 	   messages=messages,
> > > 	   timeout=10,
> > > 	   request_timeout=30,
> > > 	)
> > > 	rsp = json.loads(json.dumps(rsp))
> > > 	content = rsp['choices'][0]['message']['content']
> > > 	rel_text = content
> > > 	print(rel_text)
> > > except Exception as e:
> > >     print(e.args)
> > >     time.sleep(60)
> > > ```

---

> > > ### Author Response · Authors · 2023-08-19
> > > **Response (Part three)**
> > >
> > > **Code of Visual Cues Generation**
> > > ```python
> > > example_triplets = ['carrying_product_human', 'carrying_product_animal', 'carrying_animal_animal']
> > > raw_rel_prompts_dict = {}
> > >
> > > for rel_sub_obj_key in example_triplets:
> > > 	rel, sub, obj = rel_sub_obj_key.split('_')
> > > 	prompts = '''
> > > Known: a visual triplet is formulated as [subject, predicate, object].
> > > Note that: [position] must not include nouns other than subject and object! [position] must contain [orientation: ("above", "below", "left", "right", "inside"), shape: ("horizontal", "vertical", "square"), distance: ("small distance", "mid distance", "large distance")]!
> > > Describe the visual features of the predicate "sitting on" in a photo, when subject belongs to [human], object belongs to [product]:
> > > [subject]:
> > > - with legs.
> > > - with hip.
> > > [object]:
> > > - with flat surface.
> > > [position]:
> > > - square subject above horizontal object with a small distance.
> > >
> > > Describe the visual features of the predicate "{}" in a photo,
> > > when subject belongs to [{}], object belongs to [{}]:
> > > -'''.format(rel, sub, obj)
> > > 	messages=[
> > > 	    {"role": "user", "content": prompts}
> > > 	]
> > > 	try:
> > > 	    rsp = openai.ChatCompletion.create(
> > > 	        model="gpt-3.5-turbo",
> > > 	        messages=messages,
> > > 	        timeout=10,
> > > 	        request_timeout=30,
> > > 	    )
> > > 	    rsp = json.loads(json.dumps(rsp))
> > > 	    content = rsp['choices'][0]['message']['content']
> > > 	    rel_text = content
> > > 	    print(rel_sub_obj_key, rel_text)
> > > 	    raw_rel_prompts_dict[rel_sub_obj_key] = rel_text
> > > 	except Exception as e:
> > > 	    print(e.args)
> > > 	    time.sleep(60)
> > > ```

---

> > > > ### Comment · Reviewer_EWRp · 2023-08-21
> > > >
> > > > Thanks for the detailed response.
> > > > The author's detailed reply solved my problem. After repeated code experiments and thinking, I think this article is meaningful.
> > > >
> > > > However, I still want to emphasize that this article received a lot of negative scores at the beginning. It has a lot of shortcomings in the clarification of technical details, the presentation of specific motivation, and the exhibition of experimental performance. In order to understand the core idea of ​​the author, lengthy repeated discussions are required. It is not friendly for the readers in our community.
> > > >
> > > > Based on the above comments, I consider improving my score, but I hope that the author will add the new experimental evidence discussed with the reviewers to the final version of the paper.

---

> > > > > ### Author Response · Authors · 2023-08-21
> > > > >
> > > > > We sincerely appreciate your decision to raise the score after reviewing our rebuttal. We are delighted that you found our work is meaningful. We will add the new experimental evidence discussed with the reviewers and further polish our presentation in the new manuscript.

---

### Official Review · Reviewer_4NUf · 2023-07-05

**Soundness:** 2 fair
**Presentation:** 2 fair
**Contribution:** 3 good
**Rating:** 5
**Confidence:** 4

**Summary:**

This paper aims to address the VRD problem using LLMs. The paper decomposes the visual features into human, object, and spatial features, and designs prompts to generate visual cues that describe each of these types of visual features. The relation classification is established by calculating the distance between visual and semantic features, and dynamic weights generated by LLMs are also integrated to enhance the training process.

**Strengths:**

1. This paper focuses on a critical issue in visual relationship detection tasks, exploring the potential of leveraging LLMs to enhance visual relationship understanding.
2. The paper introduces a novel and reasonable approach by decomposing each predicate category into human, object, and spatial descriptions.
3. The authors thoroughly investigate various approaches to enhance the quality of prompts, encompassing both the generation of visual cues and the improvement of weights.

**Weaknesses:**

1. It appears that the visual cues employed in the main papers are presented as mere examples, leaving uncertainty regarding the specific visual cues utilized in the experiments. Furthermore, the visual cues depicted in Figures 3 and 4 exhibit notable differences, with Figure 4 generating more complex sentences. As a result, evaluating the quality of the visual cues based on the current evidence becomes challenging.
2. The RECODE's performance gain on the HICO-DET and V-COCO datasets is marginal, and the authors did not provide error bars in their reports. Additionally, the ablation studies were solely conducted on the VG dataset, which could have substantial differences compared to the HICO-DET dataset. Consequently, it is difficult to be convinced that the RECODE is as effective as claimed by the authors.
3. The main technical contribution of this paper lies in the development of specifically designed prompts. However, the improvements made to the prompts are relatively straightforward, and the utilization of CoT is a standard practice.
4. There are many incurious statements/claims. For example, in line 47-55, it is unclear why a person has to stand while holding an object.; in lines 67-69, it is not clarified why the act of holding depends on spatial factors.

**Questions:**

1. Could the authors provide additional examples of prompts and the corresponding visual cues generated by GPT that were utilized in the real experiments?
2. Could the authors present more empirical evidence to further support the benefits of RECODE?

**Limitations:**

The authors have adequately discussed the limitations.

---

> ### Author Rebuttal · Authors · 2023-08-09
>
> Thank you for the detailed comments. We are willing to address all your questions.
> ## Clarification of Visual Cues
>
> - **Examples in Figure 3 and Figure 4.**
> The short prompts showcased in Figure 4 were primarily intended to demonstrate the enhanced accuracy achieved by the Guided Relation Component Description. In the real experiment, to standardize the output format, we carried out a **normalized visual cue description prompt** (cf, **Section B** in supplementary materials), as mentioned in footnote2 in Line 174. All of our experiments were conducted using these normalized prompts to maintain uniformity in our approach. We will revise this part to reduce misunderstanding.
>
> - **Extra visual cue examples.**
> We would like to draw your attention to the fact that Figure 2, Figure 6, and Figure 7 already showcase a selection of generated visual cues derived from the normalized visual cue description prompt. Besides, Fig\._R 2 (cf, **PDF file**) also shows extra visual cues generated by the normalized prompt.
>
> ## Empirical Evidence
>
> - **Error Bars.**
> As shown in Fig\._R 3 (cf, **PDF file**), we plotted the error bars for three splits on the HICO-DET dataset. Across the three distinct splits - Full, Rare, and Non-Rare - our model's performance showcases impressive stability, i.e., 32.5% to 32.8%, 33.18% to 33.33%, and 32.2% to 32.55% for the Full, Rare, and Non-Rare split, respectively. These minuscule error bars reflect the high degree of consistency in our results, indicating that the measured values are reliable and repeatable.
>
> - **Ablation studies on HICO-DET and VCOCO datasets.** We have extended our ablation studies to encompass both the HICO-DET and VCOCO datasets. The results of these ablation studies are presented in Table_R 3, providing a more comprehensive evaluation of the effectiveness and generalizability of our proposed RECODE method.
>
> **Table_R 3**: Ablation studies on the HICO-DET and VCOCO datasets.
> | Cue | Spatial | Weight | HICO-DET (Full) | HICO-DET (Rare) | HICO-DET (Non-Rare) | VCOCO (Scenario 1) | VCOCO (Scenario 2) |
> |---|---|---|---|---|---|---|---|
> |   |   |   | 30.9 | 30.7 | 31.0 | 25.5 | 28.6 |
> |  ✔️ |   |   | 32.5 | 33.0 | 32.2 | 25.8 | 28.9 |
> |  ✔️ | ✔️ |   | 32.6 | 33.0 | 32.4 | 25.7 | 28.8 |
> | ✔️ |   |✔️ | 32.7 | 33.1 | 32.5 | 25.9 | 29.0 |
> | ✔️ | ✔️ | ✔️ | **32.7** | **33.2** | **32.5** | **26.0** | **29.0** |
>
>
>
> - **More comparisons with SOTA methods.** In Table_R 1 and Table_R 2, we also reported the results of combining different SOTA visual-language models and comparing with other SOTA SGG methods, which proves the effectiveness and universality of our method.
>
> ## Contributions
> Thank you for your thoughtful feedback on the main technical contribution of our paper. We appreciate the opportunity to clarify our focus and emphasize the key aspects of our work (cf, the **global response** for main contributions).
>
> - We point out that the challenges of relation sensitivity, spatial discriminability, and computational efficiency in zero-shot VRD. We achieve this by decomposing the task into three distinct components: subject, object, and spatial descriptions. Designing specific prompts is a feasible way to achieve this goal, which is not the central focus of our contribution.
>
> - We claim that the importance of each component description is different. Notably, we are the first work to utilize LLMs to effectively assign reasonable weights to these components. The CoT is employed solely to enhance the rationality of the weight generation process.
>
> ## "Incurious" Statements
> Regarding the statement in Lines 47-55, we sincerely apologize for any confusion caused. Our intention was not to assert an absolute requirement for a person to stand while holding an object. Rather, we aimed to convey that in many observed scenarios, individuals are more commonly seen standing while holding objects, and we used the phrase "**might be**" in Lines 47-55 to indicate this likelihood. Similarly, we appreciate the need for clarity in the statement made in Lines 67-69. We would like to emphasize that spatial cues can play a crucial role in distinguishing certain relations, such as "laying on" and "holding," where subject and object cues alone may not be sufficient for discrimination. We intend to revise this statement to better reflect our intent, underscoring the importance of spatial cues in the identification of specific relations.

---

> > ### Comment · Reviewer_4NUf · 2023-08-15
> >
> > Thanks for the author's rebuttal. It has addressed most of my questions. However, the contribution in the technical side is still believed to be limited as stated in my previous review, while I acknowledge the introduction of LLM is interesting and effective. Based on such considerations, I’d like to recommend a borderline accept and suggest the final version to incorporate those clarifications and results stated in the rebuttal.

---

> > > ### Author Response · Authors · 2023-08-15
> > >
> > > Thank you for raising score! We will incorporate the clarifications and results in the revised version. If you have any further questions or concerns, welcome to discuss with us. Your feedback is greatly appreciated!

---

### Official Review · Reviewer_q7Rt · 2023-07-07

**Soundness:** 2 fair
**Presentation:** 3 good
**Contribution:** 3 good
**Rating:** 5
**Confidence:** 5

**Summary:**

This paper proposed a novel method for zero-shot visual relation detection by leveraging LLM (e.g. GPT) and VLM (e.g. CLIP). Specifically, the proposed approach decomposes each predicate category into subject, object and spatial component and enrich each section with the help of LLMs, which can generate the description-based visual cues to help distinguish semantically similar concepts. Different visual cues are used to enhance discriminability from different perspectives, and the authors again use LLM to assign weights to different components for effective fusion. Extensive experiments on four different datasets are provided to demonstrate the effectiveness and interpretability.

**Strengths:**

1. The proposed approach is theoretically sound and intuitive. Enriching the prompt from class-based to description-based can provide more information to enhance the relation sensitivity, and it also improves the explainability as the relation classification score can reveal the most important factors for the prediction.
2. The decomposition of subject-object pair makes it much more efficient for processing visual signals as the previous O(N^2) patches now reduce to O(N). The spatial relationship also makes sense as an abstract from real objects to just the relations.
3. The paper is well-written and easy to follow. The extensive experiments and ablation studies/visualization help a lot for understanding the model.

**Weaknesses:**

1. The most important issues with this paper is that the evaluation section does not have important baselines. Specifically, in Table 1 and Table 2 the authors only show the performance of the proposed model with simplified version (CLS and CLSDE), which more like an ablation study. Many previous work actually attempted similar tasks and have been experimenting on the same dataset, e.g. [1][2][3].
2. In Table 2 I guess the bolded numbers should be the highest (best)? For HICO-DET, CLS has the same performance on "Rare" category  with RECODE thus should be highlighted as well I think?
3. A very very minor issue: the zero-shot chain-of-thought prompt used in most literatures are "let's think step by step" not "let's think it step by step". Formal usage should be "think" or "think about it" or "think through it", rather than "think it".

[1] https://arxiv.org/abs/1804.10660
[2] https://arxiv.org/abs/1707.09423v2
[3] https://arxiv.org/pdf/2004.00436.pdf

**Questions:**

Overall I think the paper is solid just that the evaluation is insufficient, as it doesn't have any comparison with previous methods. I will consider raising my scores if the comparison is provided in a revised version (I only listed a few papers in the weaknesses sections and I'm pretty sure they are not the latest ones, please compare against the most recent one/SOTA methods as it is more meaningful),

**Limitations:**

This paper doesn't discuss the limitations of the proposed methods, for example one important underlying assumption is the reliability of the LLM (responsible for decomposition and weight estimation) and the quality of the VLM model. I don't see obvious potential negative societal impact with this paper.

---

> ### Author Rebuttal · Authors · 2023-08-09
>
> Thank you for the detailed comments. We are willing to address all your questions.
> ## Comparison with More Baselines
> **Table_R2**: Comparison with SOTA VRD methods on the VG dataset. Note that none of these methods can be applied in the **training-free** zero-shot setting.
>
> | Model               | No Training | Unseen Relation | Training Data Source | zR@20 | zR@50 | zR@100 |
> |---------------------|-------------|-----------------|----------------------|-------|-------|--------|
> | Motifs [1]        | ❌          | ❌              | VG                   | 8.9   | 15.2  | 18.5   |
> | COACHER [2]       | ❌          | ❌              | VG & ConceptNet      | 28.2  | 34.1  | 37.2   |
> | DPL [3]           | ❌          | ❌              | VG                   | 6.0   | 7.7   | 9.3    |
> | CaCao [4]         | ❌          |    ✔️            | VG & CC3M & COCO     | 17.2  | 21.3  | 23.1   |
> | RECODE (ours)       |    ✔️   |      ✔️        |   --                  | 8.2   | 16.1  | 23.2   |
>
> - **Comparison with Training-based Methods.** Here we compared the proposed **training-free** RECODE framework with those well-designed training-based ones. Note that such comparisons are **unfair** as training-based frameworks can learn the underlying patterns and data distribution from the training set. For completeness, we still reported the results and investigate the performance gap between training-based frameworks and RECODE. Specifically, we compared the proposed RECODE with several relevant baselines, including triplet-level zero-shot VRD [1, 2], few-shot VRD [3], and category-level zero-shot VRD [4]. Since all of them cannot detect relations without training, we reported Zero-shot Recall@K (**zR@K**), which only calculates the Recall@K for those unseen **triplet** categories.
>
>   - Triplet-level zero-shot VRD methods:
>     - Motifs [1] is a traditional strong baseline without explicitly modeling the nature of zero-shot.
>     - COACHER [2] explicitly models the nature of zero-shot and takes the power of common sense from ConceptNet, resulting in better performance.
>   - Few-shot VRD methods:
>     - DPL [3] is a few-shot baseline, which mainly investigates making predictions with a few examples (here we evaluate 1-shot).
>   - Category-level zero-shot VRD methods:
>     - CaCao [4] also explicitly models the nature of zero-shot, and leverages language information from captions of CC3M and COCO for enhanced performance.
>
>   Surprisingly, even without training, RECODE still achieves competitive results, with zR@20, zR@50, and zR@100 of 8.2%, 16.1%, and 23.2%, respectively. This signifies its potential in handling unseen categories, due to the effective visual cues and inference mechanisms.
>
> - **Generalization on More Training-free Baselines.** Furthermore, we reported the results of our method applied in different SOTA visual-language models in Table_R 1, which also proves the effectiveness of RECODE.
>
> ## Minor Error and Limitations
>
> We appreciate your suggestions and will address the issues in the revised version. Besides, we have discussed the limitations in **Section F** in Supplementary Material.
>
> [1] Neural motifs: Scene graph parsing with global context. In CVPR, 2018. \
> [2] Zero-shot scene graph relation prediction through commonsense knowledge integration. In ECML PKDD, 2021. \
> [3] Decomposed prototype learning for few-shot scene graph generation.	arXiv preprint arXiv:2303.10863, 2023. \
> [4] Visually-prompted language model for fine-grained scene graph generation in an open world. In ICCV, 2023.

---

> > ### Comment · Reviewer_q7Rt · 2023-08-18
> > **Thanks for the additional experiments!**
> >
> > I've read the rebuttal from the authors and are satisfied with the response. I don't have additional questions and will raise my rating.

---

> > > ### Author Response · Authors · 2023-08-18
> > >
> > > We sincerely appreciate your decision to raise the rating after reviewing our rebuttal. We are delighted that you found our response satisfactory. Thank you for your positive assessment of our work.

---

> ### Author Response · Authors · 2023-08-18
>
> Thank you for your careful and thorough review once again. The deadline for our discussion is approaching. If you have any further concerns or questions, welcome to discuss with us.

---

### Official Review · Reviewer_CZ8p · 2023-07-07

**Soundness:** 3 good
**Presentation:** 3 good
**Contribution:** 3 good
**Rating:** 5
**Confidence:** 3

**Summary:**

This paper presents RECODE, a novel method for zero-shot visual relation detection (VRD), designed to address the shortcomings of models like CLIP in distinguishing subtle relation categories and spatial discriminability. RECODE leverages large language models (LLMs) to generate detailed description-based prompts for each relation class component, thereby enhancing VRD performance. The authors also introduce a chain-of-thought method that breaks down the problem into smaller parts for LLMs, thereby assigning reasonable weights for each component. The effectiveness and interpretability of the method are demonstrated through experiments on four benchmark datasets.

**Strengths:**

1. The approach introduces a novel framework, called RECODE, for zero-shot VRD that addresses the limitations of traditional class-based prompts. It decomposes the visual features of a triplet into subject, object, and spatial features and generates detailed descriptions of visual cues for each relation category. The use of chain-of-thought prompting for generating reasonable weights is a unique and creative approach.

2. The approach leverages large language models (LLMs), specifically GPT-3.5-turbo and CLIP, for the generation of descriptions and similarity calculations. The use of LLMs provides a strong foundation for generating informative and accurate descriptions of visual cues. The evaluation is conducted on four benchmark datasets, and the results demonstrate significant improvements over baseline methods.

3. The paper provides clear descriptions and explanations of the proposed framework, including the visual feature decomposing, semantic feature decomposing, and relation classification steps. The process of generating descriptions of visual cues and weights using LLMs is well-described, and the chain-of-thought method is illustrated with examples. The evaluation metrics and experimental setup are clearly presented.

4. The proposed approach addresses the challenge of zero-shot VRD by improving the discriminability of similar relation categories. By incorporating specific visual cues and generating descriptions, the approach enhances the performance of relation classification. The experimental results show significant improvements over baseline methods, demonstrating the effectiveness and interpretability of the proposed approach. The approach has the potential to advance the field of VRD and contribute to applications such as image understanding, scene understanding, and human-computer interaction.

**Weaknesses:**

While the experimental results show improvements over baseline methods, the paper lacks a thorough analysis of failure cases. Understanding when and why the proposed approach fails to accurately predict relations is crucial for identifying its limitations and potential areas of improvement. Analyzing failure cases and providing insights into the challenges faced by the model would strengthen the evaluation and guide future research directions.


**Questions:**

1. The paper mentions the incorporation of specific visual cues to improve the discriminability of related categories. Could you provide more details on how these cues were selected? What criteria were used to determine their relevance and effectiveness? Additionally, did you consider any alternative visual cues during the experimentation process? Exploring different visual cues and discussing their impact could provide further insights into the effectiveness of the proposed approach.

2. The paper demonstrates improvements in relation classification for zero-shot VRD, but it would be valuable to discuss the scalability of the proposed approach. How does the performance scale with an increasing number of related categories and visual concepts? Are there any computational or efficiency limitations that arise when dealing with larger datasets or more complex scenes?

3. The paper should provide a clear justification for the choice of evaluation metrics used to assess the performance of the proposed approach. Are there any limitations or biases associated with the selected metrics? Additionally, it would be helpful to include a discussion on the limitations of these metrics in capturing the true performance of zero-shot VRD models.

4. While the paper discusses the experimental results and improvements over baseline methods, it would be valuable to have a section dedicated to the limitations of the proposed approach. Identifying and addressing these limitations can help guide future research directions. Additionally, it would be beneficial to have a discussion on potential extensions or improvements to the proposed approach that could further enhance its performance or broaden its applicability.

**Limitations:**

1. Auhors has not included the section or discussion on limitations and, if applicable, the potential negative social impact of their work.

---

> ### Author Rebuttal · Authors · 2023-08-09
>
> Thank you for the detailed comments. We are willing to address all your questions.
> ## Analysis of Failure Cases
> We sincerely appreciate the reviewer's valuable feedback and the suggestion to include a thorough analysis of failure cases in our paper. We have conducted an in-depth examination of failure cases in our proposed method. As shown in Fig\._R 2 (cf, **PDF file**), the descriptions of mismatch relations are consistent with the content of images (e.g., the armrests of the toilet), leading to mismatch yet *reasonable* predictions. We can observe certain scenarios where the relation between the subject and object could be interpreted in multiple ways, and both interpretations could be considered reasonable. For instance, in the case of the "girl-toilet", both "using" and "sitting on" are plausible given the context. However, the ground truth only contains one correct label. On the other hand, we also found instances where our method outperformed the ground truth annotations. For instance, in the case of "man-snow", our method accurately predicted the relation as "lying on" in Fig\._R 2. This observation highlights the robustness of our approach.
>
> ## Visual Cues Selection and Alternative
> In our work, we focus on a **training-free** setting (cf, the **global response** for clarifying experimental settings), which directly involves inference for zero-shot VRD. Due to this difficult setting, there are two challenges: 1) we are unable to learn visual cues by designing training objectives. 2) we are unable to evaluate the quality of visual cues rigorously.
>
> - **Cues Selection.** To tackle these issues, we propose a specific approach for selecting visual cues, as outlined in Eq\.(1). Specifically, for the subject, object, and spatial descriptions, we leverage the power of the GPT model to generate a random number of descriptions. Subsequently, we compute the similarity between these descriptions and their corresponding visual features, ultimately taking the mean of these similarity scores. Additionally, the weights assigned to the subject, object, and spatial location components are determined by the GPT model, which is detailed in Section 2.2.2.
>
> - **Alternative Visual Cues.** In our experiments, we have thoroughly considered the use of different prompts for generating visual cues, as explained in Section 2.2.1: (1) **Relation class description** (cf, Figure 4(a)) generates descriptions for each relation class directly. (2) **Relation component description** (cf, Figure 4(b)) generates descriptions for each component of the relation separately. (3) **Guided relation component description** (cf, Figure 4(c)) incorporates high-level object category to guide generation process. As shown in Table 1, our results demonstrate that the guided relation component description yields superior performance.
>
> ## Performance Scale and Computational Limitations
> In our proposed approach, the absence of a training phase makes the overall time consumption mainly driven by the inference process. As the number of related categories increases, the total visual descriptions over all relations also increases, the similarity computation between visual features and text features of these descriptions will contribute to the increase in time consumption (cf, Table\_R 5). However, given all categories, the time consumption of similarity calculation is also unavoidable for other zero-shot work.
>
> ## Evaluation Metrics
> The evaluation metrics used in our work, such as Recall@K (R@K) and Mean Recall@K (mR@K) for SGG datasets (VG and GQA) [1] and mean Average Precision (mAP) for HOI datasets (HOI-DET and V-COCO) [2], are **widely adopted in the field of (zero-shot) visual relation detection**. However, we acknowledge that there may be certain limitations or biases associated with these metrics. Specifically, due to the influence of long-tail distribution in the datasets, the R@K may exhibit a bias towards the head predicate categories (categories with many samples) [1]. This bias could potentially affect the overall evaluation results and may not fully capture the model's performance on rare or less frequent relation categories. Thus, we also report m@R as a reference. Although these metrics are already widely used, designing good evaluation metrics for VRD itself is still an open problem.
>
> ## Limitations
> We have discussed the limitations in **Section F** in Supplementary Material.
>
> [1] Unbiased scene graph generation from biased training. In CVPR, 2020. \
> [2] Exploring structure-aware transformer over interaction proposals for human-object interaction detection. In CVPR, 2022.

---

> > ### Comment · Reviewer_CZ8p · 2023-08-17
> >
> > The authors have addressed the majority of my concerns. After evaluating their responses and taking into account feedback from other reviewers, I have chosen to uphold my original score.

---

> > > ### Author Response · Authors · 2023-08-18
> > >
> > > Thank you for the positive rating and for your thorough consideration of our responses. If you have any further questions or concerns, please don't hesitate to discuss them with us.

---

### Official Review · Reviewer_iRhq · 2023-07-07

**Soundness:** 4 excellent
**Presentation:** 4 excellent
**Contribution:** 4 excellent
**Rating:** 6
**Confidence:** 4

**Summary:**

Naively utilizing CLIP with prevalent class-based prompts for zero-shot VRD has several weaknesses, e.g., it struggles to distinguish between fine-grained relation types and neglects essential spatial information of two objects. To this end, the authors propose a novel method for zero-shot VRD: RECODE, which solves RElation detection via COmposite DEscription prompts. Specifically, RECODE first decomposes each predicate category into subject, object, and spatial components. Then, it leverages large language models (LLMs) to generate description-based prompts (or visual cues) for each component. Different visual cues enhance the discriminability of similar relation categories from different perspectives, boosting performance in VRD. To dynamically fuse different cues, they introduce a chain-of-thought method that prompts LLMs to generate reasonable weights for different visual cues.

**Strengths:**

- The framework for decomposing visual cues and using LLM to separately generate prompts for subject, object, and spatial features seems novel.
- The proposed method shows noticeable performance improvements, and the authors provided an ablation study to solidly analyze the design choices of the proposed method.

**Weaknesses:**

- The baselines in the experiments seem weak. Are the baseline methods recent enough models? To verify the effectiveness of the proposed method, the RECODE should be attached to the recent state-of-the-art model and show consistent performance improvement.

**Questions:**

Please refer to the questions in the weakness.

**Limitations:**

I cannot find a potential negative societal impact in this paper.

---

> ### Author Rebuttal · Authors · 2023-08-09
>
> Thank you for the detailed comments. We are willing to address all your questions.
>
> ## Incorporated to More Recent SOTA Models
> **Table_R 1**: Performance of combining with different SOTA pre-trained visual-language models on VG dataset. CLS$^\star$ denotes the model uses class-based prompts to compute the training-free zero-shot similarity between the image and text.
>
> | Backbone                   | Method             | R@20  | R@50  | R@100 | mR@20 | mR@50 | mR@100 |
> |--------------------------  |------------------  |------ |------ |-------|-------|-------|--------|
> | MS-CLIP [1]              | Baseline (CLS$^\star$) | 8.2   | 15.1  | 21.5  | 7.9   | 16.4  | 22.4   |
> |                           | RECODE$^\star$     | **9.2** | **17.3** | **24.7** | **8.3** | 15.4  | **22.6** |
> | DECLIP [2]              | Baseline (CLS$^\star$) | 11.0  | 18.3  | 24.4  | 11.0  | 19.0  | 27.1   |
> |                           | RECODE$^\star$     | **11.4** | **19.3** | **25.9** | 10.5  | **19.5** | **27.8** |
>
> We acknowledge the importance of comparing our proposed approach with the recent state-of-the-art (SOTA) models in the field. Since our methods are training-free (cf, the **global response** for clarifying experimental settings), we combined our methods with other SOTA pre-trained visual-language (VL) models, e.g., MS-CLIP [1] and DECLIP [2], which can achieve training-free zero-shot VRD. The results are reported in Table\_R 1. Notably, when combining RECODE$^\star$ with these different SOTA VL models, we also observed considerable performance gains compared to the baseline CLS$^\star$. These consistent improvements underline the effectiveness and generalizability of our RECODE.
>
> [1] Learning visual representation from modality-shared contrastive language-image pre- training. In ECCV, 2022. \
> [2] Supervision exists everywhere: A data efficient contrastive language-image pre-training paradigm. In ICLR, 2022.

---

> ### Comment · Reviewer_iRhq · 2023-08-17
>
> The authors' response answered my question.
> Therefore, I will keep my score for this paper.

---

> > ### Author Response · Authors · 2023-08-18
> >
> > Thank you for reviewing our paper and indicating that our response addressed your questions. We are grateful for your decision to maintain your score for the paper. If you have any further questions or feedback, please do not hesitate to get in touch.

---

### Author Rebuttal · Authors · 2023-08-09

We appreciate the feedback from all reviewers. First of all, we would like to clarify and highlight our **different experimental settings** and **main contributions** over the existing work. Then, we will address all mentioned misunderstandings or questions from each reviewer individually.

## Different Experimental Settings

- **Existing Zero-Shot Settings.** For the Visual Relation Detection (VRD) task (or Scene Graph Generation, SGG), there are several different "zero-shot" evaluation settings. More specifically, let $\mathcal{O}^{train}$ ($\mathcal{O}^{test}$) and $\mathcal{R}^{train}$ ($\mathcal{R}^{test}$) be the set of object categories and relation categories during the training (test) stage, respectively. Meanwhile, we use $\mathcal{T}^{train}$ and $\mathcal{T}^{test}$ to denote relation triplet categories, which are the combinations of object and relation categories (e.g., "man-riding-bike") in the training and test set, respectively. Currently, all existing "zero-shot" VRD/SGG work can further be categorized into two types:

    - **Triplet-level Zero-shot VRD (with training)** [1]**.** In this setting, object and relation categories remain consistent across training and inference, i.e., $\mathcal{O}^{train} = \mathcal{O}^{test}$ and $\mathcal{R}^{train} = \mathcal{R}^{test}$, while certain triplet categories remain unseen by the model in the test set, i.e., $\mathcal{T}^{train} \ne \mathcal{T}^{test}$. Since the model has acquired knowledge about all objects and relations, the emphasis lies in evaluating its ability to generalize to these novel triplets within $\mathcal{T}^{test}$.
    - **Category-level Zero-shot VRD (with training)** [2, 3]**.** Here, both object and relation categories differ between training and inference phases, i.e., $\mathcal{O}^{train} \neq \mathcal{O}^{test}$ and $\mathcal{R}^{train} \neq \mathcal{R}^{test}$. In addition, the triplet categories are also different, i.e., $\mathcal{T}^{train} \neq \mathcal{T}^{test}$. To detect objects and relations from a much larger and potentially unlimited set of possible categories, the model should learn distinguishable and generalizable knowledge from the limited training set.

- **Our Training-free Zero-Shot Setting.** As above-mentioned, all existing zero-shot VRD/SGG work still *needs a training set for parameter learning*. In this work, we focus on a more challenging setting: **training-free zero-shot VRD, i.e., it solves VRD without any training stage**. Recently, the training-free paradigm has ushered in a new era to our community [4, 5, 6]. As for VRD, this new and challenging setting can notably reduce labor costs, particularly considering the complexities of manual labeling for relations [7]. On the other hand, it poses considerable challenges to perform such hard tasks without fine-tuning and labeled data.

The differences between our new "zero-shot" setting and existing work are also illustrated in Fig\._R 1 (cf, **PDF file**).

## Main Contributions

We understand the emphasis on the significance of our contributions might not have been sufficiently highlighted in the paper. We apologize for any confusion and would like to reiterate the main focus of our work:

- **Compositional Visual Cues for VRD**: Our primary contribution lies in the utilization of compositional visual cues to facilitate the challenging task of training-free zero-shot VRD. Instead of solely focusing on designing prompts to generate better descriptions or visual cues, we proposed the RECODE method, which leverages large language models (LLMs) to generate detailed and informative descriptions for different components of relation categories, such as subject, object, and spatial cues. These descriptions serve as description-based prompts that assist the vision-language pretrained models (e.g., CLIP) in distinguishing between similar relation categories and improving VRD performance.

- **Weight Assignment with LLMs and CoT**: In our work, we explored the importance of different components in relation categories, and recognized that their contributions are not equal. To address this, we introduced a novel approach using LLMs to assign reasonable weights for each component. The chain-of-thought (CoT) method was introduced as a guidance to generate rationales and weights, making the weight assignment more interpretable and reasonable. Note that CoT is not the main focus of our work; instead, it was employed as a tool to improve the quality and robustness of weight assignment.

We hope that these clarifications will better communicate the significance and novelty of our work. We thank the reviewers once again for their valuable feedback, which has helped us improve the quality and clarity of our paper.

[1] Zero-shot scene graph relation prediction through commonsense knowledge integration. In ECML PKDD, 2021.\
[2] Compositional prompt tuning with motion cues for open-vocabulary video relation detec- tion. In ICLR, 2023.\
[3] Visually-prompted language model for fine-grained scene graph generation in an open world. In ICCV, 2023.\
[4] Visual programming: Compositional visual reasoning without training. In CVPR, 2023.\
[5] Navgpt: Explicit reasoning in vision-and-language navigation with large language models. arXiv preprint arXiv:2305.16986, 2023. \
[6] Segment anything meets point tracking. arXiv preprint arXiv:2307.01197, 2023. \
[7] The devil is in the labels: Noisy label correction for robust scene graph generation. In CVPR, 2022.

---

### Decision · Program_Chairs · 2023-09-21

**Decision:**

Accept (poster)

**Comment:**

This paper was reviewed by five experts in the field. The authors' rebuttal successfully resolved most of the concerns. Reviewers found the proposed method interesting and liked the presentation of the paper.

The AC agrees with the reviewers' assessments and believes the paper provides an interesting application of LLMs in-context learning to tackle a complex visual relationship detection (VRD) problem. Although the AC shares the major concern about weak baseline and limited improvements on existing VRD datasets HICO-DET and V-COCO, the overall novelty outweighs the concerns. The decision is to recommend the paper for acceptance. The reviewers did raise some valuable suggestions in the discussion that should be incorporated in the final camera-ready version of the paper. The authors are encouraged to make the necessary changes to the best of their ability.

Why not spotlight / oral: 1) The proposed method is interesting but a straightforward application of LLMs to VRD; 2) the VRD performance is not strong enough on popular VRD datasets HICO-DET and V-COCO.